# Contrastive Label-Embedding Alignment for Zero-Shot Text Classification

## Abstract

Zero-shot text classification (ZSC) seeks to assign texts to label spaces without relying on task-specific labeled documents. Yet, practical deployments of embedding models for classification often fall back on training task-specific classifiers (e.g., linear probes on frozen embeddings) to recover task-specific performance, reintroducing annotation costs and undermining the zero-shot setting. We introduce *contrastive label-embedding alignment*, a simple, compute-efficient alternative that uses only a handful of natural-language descriptions per label and no labeled documents. We lightly fine-tune a base embedding model so that label verbalizers and their descriptions are aligned in a shared space: a symmetric multi-positive contrastive objective pulls each verbalizer toward its associated descriptions while pushing it away from others, capturing the many-to-one label–description relation. Across four benchmarks (topic, sentiment, intent, emotion) and ten encoders (22M-600M parameters), as few as five descriptions per label yield consistent gains, improving macro-F1 by **+0.09** on average over zero-shot baselines, corresponding to relative improvements of roughly **5-13%** across models. Compared to a few-shot SetFit baseline with 8 labeled examples per class, our method attains higher mean performance with substantially lower variance across repeated runs, indicating improved stability in low-data regimes. The method uses label descriptions as the sole supervision signal to learn a label-specific embedding geometry for an off-the-shelf dual encoder via a symmetric multi-positive contrastive objective, while retaining efficient pre-encodable dual-encoder inference at test time.

## 1 Introduction

Text classification remains a central task in Natural Language Processing (NLP), supporting a wide range of applications such as sentiment analysis across domains, topic categorization of diverse document types, and intent detection in dialogue systems (Maas et al., 2011; Zhang et al., 2015b; Coucke et al., 2018; Larson et al., 2019; Sebastiani, 2002; Aggarwal & Zhai, 2012). Formally, the objective is to assign one or more labels from a predefined set to each text sample using only the information contained in the text itself. While progress in supervised learning has led to substantial improvements in classification accuracy, these approaches rely on large-scale, high-quality annotated datasets. Constructing such datasets is often prohibitively expensive and time-consuming, particularly in specialized domains where expert annotation is required (Settles, 2012; Ratner et al., 2017).

Zero-shot text classification (ZSC) has emerged as a compelling alternative, enabling models to assign labels that were not observed during training (Yin et al., 2019). ZSC methods exploit the semantic relationships between input texts and candidate labels, typically leveraging pretrained language models that encode these relationships based on extensive pretraining over large corpora (Brown et al., 2020; Liu et al., 2023). A widely adopted approach is to prompt large language models (LLMs) with the input text and candidate label verbalizers, allowing the model to rank or score each label. While effective, this strategy incurs considerable computational cost and latency, limiting its practicality for large-scale or real-time applications (Brown et al., 2020; Schick & Schütze, 2021; Liu et al., 2023).

Concurrently, text embedding models have seen substantial progress (Reimers & Gurevych, 2019; Gao et al., 2021; Muennighoff et al., 2023). These models map textual inputs to dense vector spaces, positioning semantically similar texts close together. This structure enables efficient similarity-based retrieval and, in principle, supports zero-shot classification by embedding both input texts and candidate label representations into a shared space and applying nearest-neighbor matching (Reimers & Gurevych, 2019; Gao et al., 2021; Fei et al., 2022). However, while such zero-shot approaches are theoretically feasible, their performance in practice is often limited, especially on challenging or fine-grained classification tasks. As a result, it is common to further adapt embedding models for classification by training a linear probe or classifier head using labeled data (Neelakantan et al., 2022; Enevoldsen et al., 2025; Chung et al., 2025), thereby reintroducing the need for annotated resources and undermining the zero-shot premise.

A parallel strand of research leverages external language and knowledge resources, including dictionary-style definitions, encyclopedic entries such as Wikipedia, and lexical ontologies such as WordNet, to provide semantic structure for zero-shot or "dataless" text classification. Early work introduced lexical resources to enrich text representations and label semantics ((Miller, 1995), see also (Scott & Matwin, 1998)), while Wikipedia-based methods mapped texts and labels into concept spaces using explicit semantic representations (Gabrilovich & Markovitch, 2007) and later demonstrated gains in downstream classification (Wang et al., 2009). More generally, dataless classification methods formalized how labels and documents can be compared via semantic proxies rather than task-specific annotations (Chang et al., 2008), and subsequent approaches operationalized label names and short natural-language descriptions as supervision signals for improved zero-shot performance (Gao et al., 2023; Chai et al., 2020; Meng et al., 2020).

Building on these insights, we propose *contrastive label-embedding alignment*, a description-based supervision framework tailored to dual-encoder text embedding models in the zero-shot setting. Rather than relying on labeled documents together with task-specific classifier heads, which both require costly annotation and additional training, our method uses a small set of natural-language descriptions per label as the *sole* supervision signal. We embed both label verbalizers and descriptions with a shared dual encoder and train it with a symmetric multi-positive contrastive objective that pulls each verbalizer toward its associated descriptions while pushing it away from descriptions of other labels, thereby inducing a label-specific embedding geometry. At test time, documents and label representations can be pre-encoded once and compared via simple similarity search, preserving the computational advantages of dual-encoder inference while leveraging description-based supervision to sharpen decision boundaries in the embedding space. Our formulation is inspired by foundational work in contrastive learning such as DrLIM, InfoNCE, SimCLR, and CLIP, but adapts these ideas to the alignment of textual label verbalizers with natural-language descriptions (Hadsell et al., 2006; van den Oord et al., 2018; Chen et al., 2020a; Radford et al., 2021b).

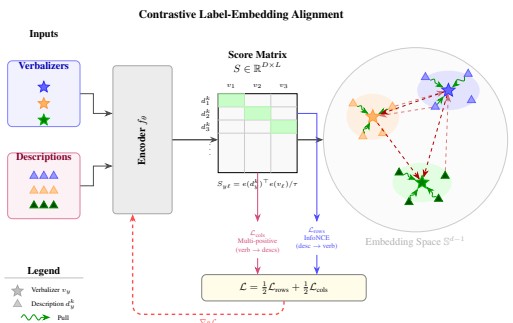

Figure 1: **Contrastive label-embedding alignment.** Label verbalizers (★) and their natural-language descriptions (▲) are encoded by a shared text encoder $f_\theta$, producing a similarity matrix $S \in \mathbb{R}^{D \times L}$ over all $D$ descriptions and $L$ verbalizers (green blocks indicate positive pairs). A *rowwise* InfoNCE loss pulls each description toward its own verbalizer, while a *columnwise* multi-positive loss aggregates each verbalizer toward all its descriptions. In embedding space $\mathbb{S}^{d-1}$, this yields tighter within-class clusters and larger inter-class margins; gradients $\nabla_\theta \mathcal{L}$ update only the encoder and require no labeled documents.

## 2 RELATED WORK

**Zero-shot and "dataless" text classification.** Early research in dataless classification replaced labeled data with semantic proxies such as label names, seed words, or external knowledge bases (e.g., WordNet, Wikipedia), enabling documents and labels to be compared in a shared semantic

space (Miller, 1995; Scott & Matwin, 1998; Gabrilovich & Markovitch, 2007; Chang et al., 2008; Wang et al., 2009). More recent methods frame ZSC as textual entailment between input texts and label verbalizers, often leveraging pretrained language models to provide the entailment signal (Yin et al., 2019). Another line explores natural-language label descriptions (e.g., definitions or short summaries) as supervision, showing improved robustness and transfer across domains (Chai et al., 2020; Meng et al., 2020; Gao et al., 2023). Despite these advances, most approaches rely on cross-encoder architectures, which require jointly encoding each document with every candidate label at inference. This results in inference costs that scale linearly with the number of labels and prevents caching of document embeddings, making such methods impractical for large label sets or real-time deployment.

**Few-shot learning.** Few-shot methods fine-tune compact encoders on small labeled sets, bridging the gap between zero-shot and fully supervised learning. SetFit exemplifies this paradigm in the context of dual-encoder text embedding models: it first fine-tunes an embedding model contrastively and then trains a lightweight classifier head on top of the resulting embeddings, achieving strong results with limited supervision and modest compute (Tunstall et al., 2022). On the other hand, parameter-efficient fine-tuning techniques (e.g., adapters, LoRA) reduce the compute cost of task-specific training by limiting the number of updated parameters, but they do not alleviate the central bottleneck of acquiring labeled examples (Houlsby et al., 2019; Hu et al., 2022).

**In-context learning with large models.** Large language models (LLMs) can perform zero- or few-shot classification via in-context learning (ICL), where label names and demonstration examples are provided directly in the prompt (Dong et al., 2024; Luo et al., 2024). While often effective out of the box, ICL comes with several challenges. Performance can be highly sensitive to the selection and ordering of demonstrations, and although recent models support much larger context windows, reliably exploiting long prompts remains difficult due to dilution and context "forgetting." Moreover, it is non-trivial to enforce consistently parseable, deterministic label outputs, which complicates downstream use. Finally, inference is computationally expensive, as the model must process the entire prompt together with each new example. In light of these issues, comparisons indicate that fine-tuned encoders can offer more stable and compute-efficient behavior for sustained deployment on targeted tasks (Mosbach et al., 2023).

**Embedding models for classification and related tasks.** Recent sentence and document embedding models trained with large-scale contrastive or instruction-tuning objectives (e.g., SBERT, SimCSE, E5, GTE, BGE, EmbeddingGemma, Qwen3-Embedding) provide strong transfer across retrieval, semantic similarity, clustering, and text classification benchmarks (Reimers & Gurevych, 2019; Gao et al., 2021; Wang et al., 2022; Li et al., 2023; Xiao et al., 2023; Google DeepMind & Google Research, 2025; Zhang et al., 2025). These dual-encoder architectures independently encode inputs into a shared vector space, enabling efficient nearest-neighbor search and scalable deployment. For classification, a common strategy is to train linear probes or lightweight classifier heads on top of frozen embeddings (Neelakantan et al., 2022; Muennighoff et al., 2023; Enevoldsen et al., 2025), while retrieval and semantic similarity tasks are typically handled via direct similarity scoring. In principle, the same machinery can support zero-shot classification by comparing document embeddings to label representations, but naïve instantiations often struggle on many tasks, highlighting the need for better alignment between label semantics and the embedding space.

**Contrastive learning.** Contrastive learning objectives such as InfoNCE and SimCLR-style losses encourage representations of semantically related views to be close in embedding space while separating unrelated examples, often using large batches or memory banks to provide in-batch negatives (van den Oord et al., 2018; Chen et al., 2020a). This paradigm has proved highly effective for learning transferable representations across modalities. In vision-language settings, CLIP trains dual encoders on large-scale image-text pairs, aligning the two modalities in a shared space and enabling flexible retrieval of images from textual queries (and vice versa) via simple similarity comparisons (Radford et al., 2021b). In text-only settings, early work applied contrastive objectives to augmented or paired text views (e.g., dropout-based augmentations, paraphrases, or neighboring sentences), yielding sentence and document encoders with strong performance on semantic similarity and related tasks (Reimers & Gurevych, 2019; Gao et al., 2021). Subsequent models scale this recipe to much larger and more diverse corpora (e.g., E5, GTE, BGE), producing more "universal"

embedding models that transfer well and can be easily adapted to a wide range of downstream tasks, including retrieval, clustering, and classification benchmarks (Wang et al., 2022; Li et al., 2023; Xiao et al., 2023). Our work adopts this contrastive dual-encoder perspective but tailors the objective to align label verbalizers with small sets of curated natural-language descriptions, shaping the embedding space for zero-shot classification without relying on labeled documents.

## 3 CONTRASTIVE LABEL-EMBEDDING ALIGNMENT

At a high level, our goal is to reshape an off-the-shelf dual-encoder text embedding model so that label verbalizers act as clean, well-positioned "representatives" of their labels in embedding space, purely based on natural-language descriptions of what those labels mean. Figure 1 provides a visual summary. We start from a pretrained text encoder and assume access only to a small set of descriptions per label, written as short paragraphs that spell out the intended meaning and scope of each class. We then fine-tune the encoder with a contrastive objective that (i) pulls each description toward its correct label verbalizer and (ii) moves each verbalizer toward the dense region of its own description cloud while repelling it from descriptions of other labels. Concretely, we construct a description-verbalizer similarity matrix and optimize a combination of rowwise InfoNCE and a columnwise multi-positive variant that captures the many-to-one relation between labels and their descriptions.

**Setup and notation.** Let $\mathcal{Y} = \{1, \ldots, L\}$ denote the label set. For each $y \in \mathcal{Y}$ we assume two kinds of textual anchors:

- a short *verbalizer* $v_y$ (e.g., for $y =$ "Sports" we might use $v_y =$ "This news snippet is about sports."), used at inference time to represent label $y$;[1]

- a small set of *label descriptions* $\mathcal{D}_y = \{d_y^k\}_{k=1}^{K_y}$, written as short paragraphs that clarify the kinds of documents $y$ should cover.

We denote the union of all descriptions and its size by

$$\mathcal{D} = \bigcup_{y \in \mathcal{Y}} \mathcal{D}_y, \qquad D = \sum_{y \in \mathcal{Y}} K_y.$$

No labeled documents are used at training time; all supervision flows through these verbalizers and descriptions.

**Encoder and similarity scores.** We use a single encoder $f_\theta$ with its native pooling map $\pi(\cdot)$.[2] Given a text $t$ with $S$ tokens, the encoder produces contextual token representations

$$f_\theta(t) \in \mathbb{R}^{S \times d}.$$

These are pooled and $\ell_2$-normalized to obtain a sentence embedding

$$e(t) = \frac{\pi\big(f_\theta(t)\big)}{\big\|\pi(f_\theta(t))\big\|_2} \in \mathbb{R}^d,$$

so cosine similarity reduces to a dot product. With temperature $\tau > 0$, the similarity between a description $d$ and a verbalizer $v$ is

$$s(d, v) = \frac{e(d)^\top e(v)}{\tau}.$$

We reuse the encoder architecture and pooling strategy from the base model and only update $\theta$; no additional layers or task-specific heads are introduced.

---

[1]Appendix H studies variants that (i) omit the verbalizer and use the label text directly, or (ii) replace it with the mean embedding of the descriptions.

[2]We use the pooling native to the pretrained model, e.g., CLS-token, mean, or last-token pooling.

**Batch structure and anchors.** Each training batch considers the cross-product between all descriptions $\mathcal{D}$ and all verbalizers $\{v_1, \ldots, v_L\}$, forming the score matrix

$$S \in \mathbb{R}^{D \times L}, \qquad S_{y\ell}^k = s\big(d_y^k, v_\ell\big).$$

It is helpful to view this matrix from two complementary perspectives:

- *Row-anchors (descriptions).* Each row corresponds to a single description $d_y^k$ and should assign high probability to its correct label $y$ while treating other labels $\ell \neq y$ as negatives.

- *Column-anchors (verbalizers).* Each column corresponds to verbalizer $v_\ell$ and should collect probability mass from all of its positives $\{d_\ell^k\}_{k=1}^{K_\ell}$ while discounting descriptions of other labels.

This row/column duality is central: rows enforce *one-positive discrimination* (each description chooses a label), while columns implement *multi-positive aggregation* over a label's description set.

### 3.1 ROWWISE INFONCE: CLASSIFYING DESCRIPTIONS INTO LABELS

From the rowwise viewpoint, each description $d_y^k$ is a query that must identify its label $y$ among all $L$ options. The induced distribution over labels is

$$p(\ell \mid d_y^k) = \frac{\exp\{S_{y\ell}^k\}}{\sum_{j=1}^{L} \exp\{S_{yj}^k\}}.$$

The rowwise InfoNCE objective averages the cross-entropy against the correct label $y$:

$$\mathcal{L}_{\text{rows}} = \frac{1}{D} \sum_{y \in \mathcal{Y}} \sum_{k=1}^{K_y} \Big( \log \sum_{j=1}^{L} e^{S_{yj}^k} - S_{yy}^k \Big). \tag{1}$$

This is equivalent to a multiclass classifier over labels, where each description is a training example and the verbalizers serve as the representative embeddings for each class. Intuitively, equation 1 pulls each $d_y^k$ toward its own verbalizer $v_y$ while pushing it away from verbalizers $v_{\ell \neq y}$, tightening the alignment between descriptions and their labels.

### 3.2 COLUMNWISE MULTI-POSITIVE INFONCE: AGGREGATING OVER DESCRIPTION SETS

The rowwise objective treats each description independently. However, a label is not defined by a single canonical description, but by a *set* of complementary descriptions that cover different facets, edge cases, or typical failure modes. The columnwise objective explicitly models this many-to-one relation.

From the column perspective, each verbalizer $v_\ell$ has a set of positives $\mathcal{D}_\ell = \{d_\ell^k\}_{k=1}^{K_\ell}$, and all descriptions $d_y^k$ with $y \neq \ell$ are negatives. We define the global and positive-partition normalizers

$$Z_\ell = \sum_{y \in \mathcal{Y}} \sum_{k=1}^{K_y} \exp\{S_{y\ell}^k\}, \qquad Z_\ell^+ = \sum_{k=1}^{K_\ell} \exp\{S_{\ell\ell}^k\}.$$

The columnwise objective maximizes the aggregated positive mass relative to the global normalizer:

$$\mathcal{L}_{\text{cols}} = \frac{1}{L} \sum_{\ell=1}^{L} \big( \log Z_\ell - \log Z_\ell^+ \big). \tag{2}$$

This is a *set-level* multi-positive term: it optimizes the combined probability of a label's positives rather than treating them as independent single-positive examples.

The log-sum-exp structure has two important consequences:

- **Robustness to heterogeneous descriptions.** Strong, representative descriptions contribute more to $Z_\ell^+$, while noisy or idiosyncratic descriptions contribute less. Formally, the gradient

$$\frac{\partial \mathcal{L}_{\text{cols},\ell}}{\partial S_{\ell\ell}^k} = \frac{e^{S_{\ell\ell}^k}}{Z_\ell} - \frac{e^{S_{\ell\ell}^k}}{Z_\ell^+}$$

  induces *adaptive within-positive weighting* proportional to $e^{S_{\ell\ell}^k}/Z_\ell^+$, automatically down-weighting outliers and emphasizing representative descriptions.[3]

- **Stability w.r.t. description count.** Because the objective depends on the ratio $Z_\ell^+/Z_\ell$ and is normalized per label, it remains stable even when labels have different numbers of descriptions $K_\ell$.[4]

Geometrically, equation 2 pulls each $v_\ell$ toward the high-density region ("cloud") formed by its descriptions while repelling it from the description clouds of other labels.

### 3.3 Final objective and connection to standard InfoNCE

Our training loss is a simple symmetric combination of the rowwise and columnwise terms:

$$\mathcal{L} = \tfrac{1}{2}\,\mathcal{L}_{\text{rows}} + \tfrac{1}{2}\,\mathcal{L}_{\text{cols}}. \tag{3}$$

This symmetry encourages consistency in both directions: descriptions should clearly identify their label (rows), and each label should be well-explained by its description set (columns).[5]

In standard InfoNCE-style contrastive learning with paired batches, one forms a square similarity matrix whose diagonal entries are the unique positive pairs, and all off-diagonal entries act as in-batch negatives. In our setting, the score matrix $S \in \mathbb{R}^{D \times L}$ is rectangular, and positives are defined by label consistency rather than by the matrix diagonal: any cell $(d_y^k, v_\ell)$ with $y = \ell$ is a positive, while all cells with $y \neq \ell$ serve as in-batch negatives, yielding an $O(DL)$ softmax per batch. Unit-norm embeddings constrain optimization to the hypersphere, and the temperature $\tau$ controls the sharpness of the row- and column-softmax distributions. Following common practice in contrastive learning, we fix $\tau = 0.07$ (Gao et al., 2021; Chen et al., 2020b; Radford et al., 2021a).

### 3.4 Inference as dual-encoder classification

At test time, we use the encoder as a standard dual encoder for classification. Given a document $x$, we compute its embedding $e(x)$ and score labels by similarity to the verbalizers:

$$\text{score}(y \mid x) = e(x)^\top e(v_y), \qquad \hat{y} = \arg\max_{y \in \mathcal{Y}} \text{score}(y \mid x).$$

Because both documents and verbalizers can be pre-encoded and stored, classification reduces to a nearest-neighbor search over label representations. This preserves the computational advantages of dual-encoder inference by allowing labels to be reused across large corpora.

### 3.5 Geometric intuition

The combined effect of $\mathcal{L}_{\text{rows}}$ and $\mathcal{L}_{\text{cols}}$ is easiest to understand in geometric terms. Initially, verbalizers and descriptions may be scattered: verbalizers can sit off-center relative to the document clouds of their labels, and class regions may partially overlap. The rowwise term contracts each description toward its own verbalizer and expands margins to other labels, encouraging a clear mapping from descriptions to the label-specific reference representations. The columnwise term simultaneously moves each verbalizer toward the *barycenter* of its description cloud while pushing it away from descriptions of other labels, ensuring that verbalizers end up in high-density regions of the correct class.

Figure 2 illustrates this process on AGNews (Zhang et al., 2015a) using the `all-MiniLM-L6-v2` model. In the *left* panel, verbalizers (★) sit off-center relative to the document clouds, and class

---

[3]Appendix E provides an analysis of the impact of noisy descriptions.

[4]Appendix F highlights this stability empirically.

[5]Appendix D provides an empirical analysis of the loss components.

regions partially overlap. The *middle* panel depicts the learning forces: each description $d_y^k$ (▲) is pulled toward $v_y$ and pushed away from other verbalizers; each $v_y$ is pulled toward the barycenter of $\{d_y^k\}_k$ and repelled from descriptions of other labels. After optimization, the *right* panel shows verbalizers relocated near the densest part of their label's description cloud and larger inter-label margins.

Fundamentally, although training uses only verbalizers and descriptions, the shared encoder is updated and the feature space is globally reshaped. Documents with similar semantics are steered toward their label's "attractor direction," reducing within-class dispersion and increasing between-class separation. In the 2-D UMAP view, this manifests as tighter, better-separated clouds in the right panel; in the full embedding space, it translates into more robust nearest-neighbor classification based on label verbalizers.

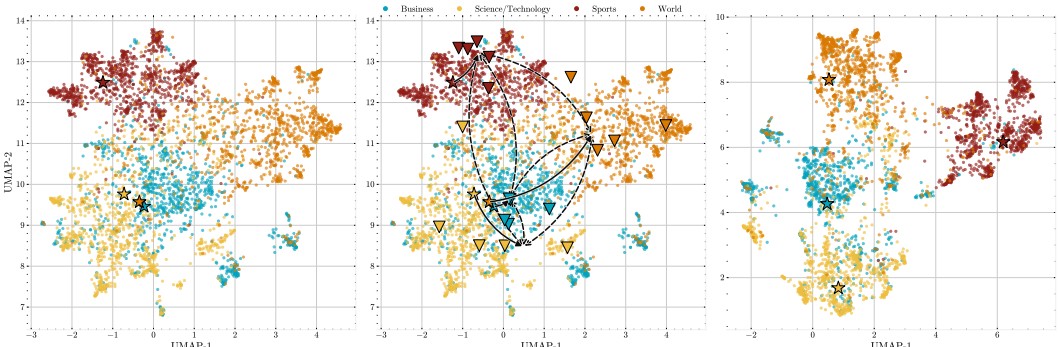

Figure 2: **AGNews (Zhang et al., 2015a).** Left: embeddings before finetuning (stars denote label verbalizers). Middle: schematic of our training forces (triangles denote label descriptions). Right: embeddings after finetuning.

## 3.6 HYPERPARAMETERS

**Batching and training length.** Because description sets are small, we treat one sweep over all description-verbalizer pairs as an *epoch*. When memory is constrained, we use gradient accumulation so that a single optimizer update corresponds to one logical sweep. We cap the maximum iterations liberally and apply early stopping on the training loss itself, a criterion that does not require labeled validation data.

**Learning rate and uniformity selection.** Performance is sensitive to the learning rate (LR). Overly aggressive LRs can trigger *representation collapse* (especially mode collapse (Bardes et al., 2021)) in our small-data regime, whereas simply reducing the LR avoids hard collapse but can stall progress and undercut alignment. We follow the view that contrastive learning balances *alignment* and *uniformity* on the hypersphere (Wang & Isola, 2020). In our setup, alignment is enforced by the supervision signal (descriptions ↔ verbalizers), so the main concern is to *preserve uniformity* so that the embedding space does not degenerate.

We therefore select the LR using a label-free uniformity criterion computed on an unlabeled pool $\mathcal{X}_u = \{x_i\}$ from the target domain. Let $z_i = e(x_i)$ be $\ell_2$-normalized embeddings and $t > 0$ a scale parameter. Define

$$\mathcal{L}_{\text{uni}}(t) \;=\; \log \mathbb{E}_{i \neq j}\Big[e^{-t\,\|z_i - z_j\|_2^2}\Big] \;\;\approx\;\; \log\left(\frac{1}{M}\sum_{m=1}^{M} e^{-t\,\|z_{i_m} - z_{j_m}\|_2^2}\right), \qquad (4)$$

where $(i_m, j_m)$ are random distinct indices from $\mathcal{X}_u$. Lower values correspond to more uniform (i.e., less collapsed) embeddings. To select the LR, we run short warmups at candidate values and choose the one that *minimizes* $\mathcal{L}_{\text{uni}}(t)$; following Wang & Isola (2020), we fix $t = 2$. This criterion is label-free, computationally inexpensive, and in practice lower values correlate with stronger downstream performance. Figure 8 in Appendix J illustrates this correlation across a range of models and datasets, with additional details provided in the appendix.

As a fallback, reusing the base model's pretraining LR provides a safe, though non-optimized, choice.[6] Figure 3(b) illustrates AGNews document embeddings from *all-MiniLM-L6-v2* at the LR chosen by this procedure; embeddings are reduced with PCA to $\mathbb{R}^3$ and projected onto the unit sphere $\mathbb{S}^2$ via $\ell_2$-normalization.[7]

# 4 EXPERIMENTAL SETUP

We evaluate on four text-classification benchmarks: topic (AGNews (Zhang et al., 2015a)), emotion (EmotionDAIR (Saravia et al., 2018)), sentiment (RottenTomatoes (Pang & Lee, 2005)), and fine-grained intent (Banking77 (Casanueva et al., 2020)). For each dataset and class, we write exactly 5 short descriptions that characterize typical documents; examples and ablations on the number of descriptions are given in Appendices A and B. For Banking77, we report main results on six card-related intents to probe fine-grained distinctions and extend the analysis to the full label space, precision, and recall in Appendix I.

We test our method on ten pretrained dual-encoder text embedding models spanning a range of architectures and sizes (roughly 22M–600M parameters); Appendix C summarizes all models.

**Training.** We use AdamW (Loshchilov & Hutter, 2019), training for at most 1000 iterations with early stopping (patience $= 10$, tolerance $\Delta = 10^{-5}$), evaluated every 10 steps. Learning rates are swept over $\{1, 3, 5\} \times \{10^{-4}, 10^{-5}, 10^{-6}\}$ and selected using the uniformity score in Eq. equation 4, computed on 50,000 document pairs from the test subset of the target domain. To improve stability in our small-data regime, we use linear warmup during the first 50% of training steps.

**Evaluation.** To ensure comparability across tasks with different label cardinalities and class balances, we use **macro $F_1$** as our primary metric, which gives equal weight to every class and is appropriate for both balanced and imbalanced multi-class settings (Sokolova & Lapalme, 2009).

| Model | AGNews | Banking77 | EmotionDAIR | RottenTomatoes | Avg |
|---|---|---|---|---|---|
| all-MiniLM-L6-v2 | 0.67 | 0.66 | 0.35 | 0.66 | 0.58 |
| *trained* | 0.79 (+0.12) | 0.90 (+0.23) | 0.43 (+0.09) | 0.70 (+0.04) | 0.70 |
| e5-base-v2 | 0.75 | 0.79 | 0.44 | 0.83 | 0.70 |
| *trained* | 0.81 (+0.05) | 0.96 (+0.17) | 0.48 (+0.05) | 0.82 (0.00) | 0.77 |
| e5-large-v2 | 0.78 | 0.79 | 0.44 | 0.85 | 0.72 |
| *trained* | 0.82 (+0.03) | **0.96** (+0.17) | 0.53 (+0.09) | 0.86 (+0.00) | 0.79 |
| bge-base-en-v1.5 | 0.63 | 0.86 | 0.42 | 0.81 | 0.68 |
| *trained* | 0.82 (+0.19) | 0.95 (+0.09) | 0.47 (+0.06) | 0.82 (+0.01) | 0.77 |
| bge-large-en-v1.5 | 0.75 | 0.84 | 0.44 | 0.82 | 0.71 |
| *trained* | 0.82 (+0.07) | 0.95 (+0.10) | 0.56 (+0.12) | 0.85 (+0.03) | 0.80 |
| gte-base-en-v1.5 | 0.73 | 0.86 | 0.44 | 0.84 | 0.72 |
| *trained* | 0.83 (+0.09) | 0.95 (+0.08) | 0.49 (+0.06) | 0.85 (+0.01) | 0.78 |
| gte-modernbert-base | 0.75 | 0.88 | 0.45 | 0.82 | 0.73 |
| *trained* | 0.80 (+0.05) | 0.94 (+0.06) | 0.49 (+0.04) | 0.84 (+0.03) | 0.77 |
| gte-large-en-v1.5 | 0.72 | 0.92 | 0.40 | 0.87 | 0.73 |
| *trained* | 0.83 (+0.11) | 0.95 (+0.03) | 0.50 (+0.10) | 0.83 (-0.04) | 0.78 |
| Qwen3-Embedding-0.6B | 0.63 | 0.88 | 0.48 | 0.76 | 0.69 |
| *trained* | **0.85** (+0.22) | 0.92 (+0.03) | 0.57 (+0.08) | **0.88** (+0.12) | 0.80 |
| embeddinggemma-300m | 0.53 | 0.80 | 0.49 | 0.64 | 0.61 |
| *trained* | 0.74 (+0.21) | 0.94 (+0.14) | **0.57** (+0.09) | 0.73 (+0.09) | 0.74 |

Table 1: Main results by model family. Each model has a base row (zero-shot) and a trained row, with F1 scores and improvements reported in percentage points. Best trained F1 per dataset is **bold**. For each model, the largest improvement is underlined. Averages are macro-averages across datasets; trained averages include the mean improvement in parentheses.

---

[6]This heuristic proved effective across many model-dataset combinations we tested.

[7]We avoid UMAP because its locality-crowding parameters can arbitrarily distort interpoint distances, making it unsuitable for objectively visualizing uniformity.

## 5 RESULTS AND ANALYSIS

We begin with the overall effect of label alignment across all models and datasets. Averaging over ten encoders and four benchmarks (Table 1), description-only training improves macro-$F_1$ from roughly 0.68 to 0.77, i.e. by about $+0.09$ absolute. This confirms that aligning label verbalizers with a compact set of semantically rich descriptions yields a consistent and robust gain in true zero-shot transfer.

The magnitude of improvement varies by dataset. On **AGNews**, the mean increase is about $+0.12$ (range $\approx +0.04$ to $+0.22$), with particularly large jumps for smaller or less specialized encoders: for example, *bge-base-en-v1.5* improves from 0.63 to 0.82, and *embeddinggemma-300m* from 0.53 to 0.74. **Banking77** shows a similarly strong average gain of about $+0.11$, with several models registering double-digit improvements (e.g., *all-MiniLM-L6-v2* from 0.66 to 0.90, *e5-base-v2* and *e5-large-v2* from 0.79 to 0.96, and *embeddinggemma-300m* from 0.80 to 0.94). For **EmotionDAIR**, the mean improvement is more modest, around $+0.07$ (typical range $+0.04$ to $+0.12$), while **RottenTomatoes** exhibits the smallest average gain (about $+0.03$), with changes spanning from slightly negative to $+0.12$ depending on the encoder. This pattern suggests that topical and intent-based tasks benefit most from description alignment, whereas emotion recognition and binary sentiment offer less headroom, especially for already strong baselines.

Turning to dataset-specific winners, different models achieve the top post-training performance. On **AGNews**, *Qwen3-Embedding-0.6B* reaches **0.85**, reflecting one of the largest single improvements in the table (+0.22). On **Banking77**, *e5-large-v2* attains **0.96** (tied with *e5-base-v2*), matching the best scores on this benchmark. For **EmotionDAIR**, *embeddinggemma-300m* achieves the highest reported macro-$F_1$ at **0.57**, and on **RottenTomatoes**, *Qwen3-Embedding-0.6B* again leads with **0.88**. Considering macro-averaged $F_1$ across all four datasets, *Qwen3-Embedding-0.6B* and *bge-large-en-v1.5* reach the highest overall performance at **0.80**, closely followed by *e5-large-v2* at **0.79** and *gte-base-en-v1.5* and *gte-large-en-v1.5* around **0.78**.

The cost-benefit profile shows that both compact and larger encoders profit, but in different regimes. Smaller models often realize the largest relative improvements: *all-MiniLM-L6-v2* gains about $+0.12$ on average (from 0.58 to 0.70), including a jump from 0.66 to 0.90 on Banking77, while *embeddinggemma-300m* improves by roughly $+0.13$ on average ($0.61 \rightarrow 0.74$), with strong gains on AGNews and Banking77. At the same time, the strongest models in the pool also benefit. *Qwen3-Embedding-0.6B* records an average improvement of about $+0.12$ and achieves the best or tied-best scores on two datasets, underlining that substantial gains are not limited to weaker encoders. Conversely, families such as E5 and GTE start from already high baselines, particularly on RottenTomatoes (0.82-0.87 before training), which naturally constrains the headroom for further improvement and leads to more modest deltas.

Family-level trends are relatively stable. The E5 models show average gains between roughly $+0.06$ and $+0.08$, BGE models between $+0.08$ and $+0.09$, and GTE models between $+0.04$ and $+0.06$. Despite these modest increments, the GTE and BGE families remain among the strongest performers in absolute terms after description-tuning. Meanwhile, Gemma and Qwen perform above expectations given their parameter counts: *embeddinggemma-300m* becomes highly competitive after tuning, and *Qwen3-Embedding-0.6B* reaches the top average score and leads on AGNews and RottenTomatoes.

The distribution of improvements also sheds light on dataset difficulty. **EmotionDAIR** stands out as the most challenging benchmark: even the best fine-tuned model reaches only **0.57** macro-$F_1$, well below the post-training levels on AGNews, Banking77, and RottenTomatoes. This suggests that emotion recognition may require not only richer descriptions but also a larger and more diverse description set per label to adequately capture subtle and context-dependent emotional cues. In contrast, **AGNews** and **Banking77** benefit most strongly from description alignment, consistent with the intuition that topical and intent semantics are well captured by concise, high-quality definitions. On **RottenTomatoes**, the degree of improvement is inversely related to the encoder's baseline quality: weaker models (e.g., MiniLM, Gemma, Qwen) gain noticeably, while the strongest encoders improve only marginally and sometimes not at all.

**Few-shot comparison.** To contextualize our zero-shot description-only alignment, we compare against SetFit (Tunstall et al., 2022), a widely used few-shot method for embedding models. SetFit

combines a contrastive pretraining stage with a lightweight classifier head. Following the original setup, we train SetFit on EmotionDAIR with 8 samples per class and repeat the procedure 20 times with different random draws of the training set. Our approach, by contrast, uses 5 descriptions per class and generates 20 variations of the descriptions. Figure 3(a) shows that our method achieves a higher average macro-F1 and, more importantly, exhibits substantially smaller variance. While SetFit can occasionally match or exceed our performance, it displays a long tail of poor outcomes, reflecting its sensitivity to the specific few-shot samples selected.

In summary, description-only finetuning yields consistent performance gains across a diverse set of encoders and tasks. The largest improvements occur on topic and intent classification, while emotion recognition remains comparatively difficult. The method is particularly appealing in low-compute settings, where smaller models realize disproportionate benefits, yet even strong off-the-shelf encoders record non-trivial positive gains, and the best-performing models after training are among the most parameter-efficient ones.

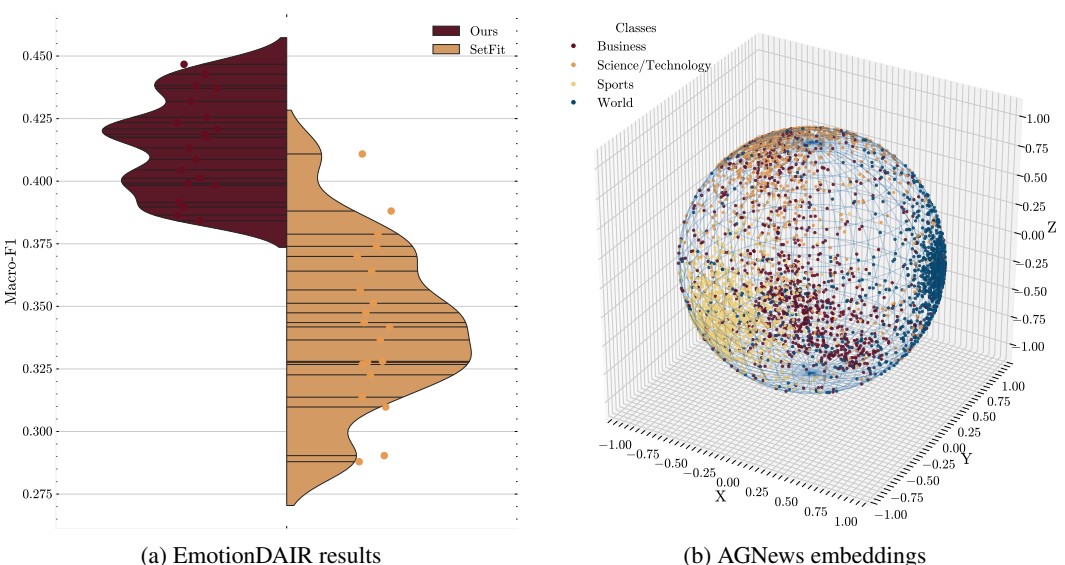

(a) EmotionDAIR results  (b) AGNews embeddings

Figure 3: (a) Performance comparison of our approach with SetFit (Tunstall et al., 2022) on the EmotionDAIR dataset (Saravia et al., 2018), across 20 runs with different sampled training sets. (b) Visualization of AGNews document embeddings after finetuning `all-MiniLM-L6-v2`, projected onto the hypersphere using PCA; colors indicate class membership.

## 6 CONCLUSION AND FUTURE WORK

We introduced *contrastive label-embedding alignment* for zero-shot text classification with dual-encoder text embedding models, using short, human-written label descriptions as the sole supervision signal. By aligning label verbalizers with their descriptions via a symmetric multi-positive contrastive objective, our method reshapes the embedding space into a label-aware geometry and yields consistent, architecture-agnostic gains over naïve zero-shot use of embeddings, averaging **+0.09** macro-F1 across ten encoders and four datasets. Compared with a few-shot SetFit pipeline using 8 labeled examples per class, it attains higher average performance with substantially lower variance across runs, while preserving efficient pre-encodable dual-encoder inference and avoiding labeled documents entirely. A natural next step is to investigate the role of hyperspherical uniformity more deeply, both in its empirical correlation with downstream performance and in ways it could be incorporated directly into the training objective.

## REPRODUCIBILITY STATEMENT

We will release the full codebase under an MIT license, including preprocessing scripts, training and evaluation routines, the uniformity-based learning-rate selection code, and all logging utilities

required to regenerate figures and tables. The label-description sets and the exact sampling protocol used for the uniformity metric will be made publicly available. Fine-tuned checkpoints for all reported models will be released in Hugging Face format, and we will document the exact pretrained encoder revisions used.

All hyperparameters (optimizer, learning-rate grids, batch sizes, gradient accumulation, early-stopping criteria) are specified in the main text at the point of use; the appendix provides additional in-depth results. Experiments were run on NVIDIA A100 80 GB GPUs, with inference carried out in `bfloat16`. We provide pinned package versions and configuration files to recreate the software environment.

We do not fix random seeds during training. Instead, we verified empirically that the results and conclusions are robust to stochasticity in initialization and sampling. We rely only on publicly available datasets and pretrained encoders, which are properly cited. To our knowledge, there are no legal or technical restrictions that would prevent exact reproduction of our results.

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

## A  DATASETS OVERVIEW

Table 2 summarizes the datasets used in our experiments, including their domains, number of classes, sources, and license terms. All datasets are publicly available via Hugging Face Datasets[8]. For label verbalizers, we follow the setup of Laurer et al. (2023).

Our selection deliberately covers four common zero-shot classification families: emotion recognition on social media (*EmotionDAIR*), intent detection in a narrow banking domain (*Banking77*), binary sentiment analysis for movie reviews (*RottenTomatoes*), and topic classification in news (*AGNews*). This yields a mix of short, informal texts (EmotionDAIR, Banking77), longer and more descriptive documents (AGNews), and highly domain-specific language (banking vs. movies vs. general news). For all datasets we use the official test splits provided by the original authors (as exposed through Hugging Face).

| Task | Domain | Dataset | # Classes | Source | License |
|------|--------|---------|-----------|--------|---------|
| Emotion | Social media | EmotionDAIR | 6 | Saravia et al. (2018) | Research/education only |
| Intent | Banking | Banking77 | 6[9] | Casanueva et al. (2020) | CC BY 4.0 |
| Sentiment | Movies | RottenTomatoes | 2 | Pang & Lee (2005) | CC0 1.0 |
| Topic | News | AGNews | 4 | Zhang et al. (2015a) | Non-commercial |

Table 2: Datasets used in the evaluation, covering emotion recognition, intent detection, sentiment analysis, and topic classification.

## B  LABEL VERBALIZERS AND DESCRIPTIONS

Our method assumes a short verbalizer and a small set of natural-language descriptions for each label. The verbalizer plays the role of a compact, sentence-level anchor (e.g., "This example news text is about world news"), while the descriptions provide richer, paragraph-level semantics that spell out the kinds of documents the label should cover. In practice, we construct these descriptions manually, starting from the label verbalizers of Laurer et al. (2023) and expanding each label into multiple complementary paraphrases. For AGNews, shown below, we write 5 descriptions per class that: (i) emphasize different aspects of the underlying topic (e.g., actors, events, context), (ii) avoid overlap with other labels (e.g., business vs. world news vs. sports), and (iii) remain generic enough to apply across articles and time periods.

These descriptions are used exclusively during our description-only training stage; no dataset examples or ground-truth labels are required to construct them. At inference time, the model receives only raw documents and the label representations (verbalizers, label names, or mean description embeddings; see Appendix H) and must classify texts in a strict zero-shot manner. Table 3 illustrates the resulting description sets for AGNews; analogous description collections are created for Banking77, EmotionDAIR, and RottenTomatoes.

---

[8] https://huggingface.co/datasets
[9] For *Banking77*, we restrict evaluation to the six card-related intent classes for fine-grained classification.

| Category (verbalizer) | Descriptions |
|---|---|
| **World News** 
 *Verbalizer:* "This example news text is about world news." | • Coverage of international affairs and geopolitics: governments, elections, diplomacy, conflicts, treaties, and sanctions. Stories focus on cross-border events and their global implications rather than domestic business or sports outcomes. 
 • News about countries interacting on the world stage: summits, UN resolutions, regional alliances, and humanitarian crises. Emphasis is on state actors, policy decisions, and shifts in international relations. 
 • Reporting on wars, ceasefires, peace talks, and military deployments across regions. Articles highlight causes, stakeholders, civilian impact, and reactions from other nations or international bodies. 
 • Global society and policy issues such as migration, human rights, climate diplomacy, and development aid. Pieces track how multiple countries respond and coordinate. 
 • International incidents and disasters (natural or man-made) where response, accountability, and cross-national coordination are central. Focus remains on worldwide context rather than local business ramifications. |
| **Sports** 
 *Verbalizer:* "This example news text is about sports." | • Results, previews, and analysis of professional or amateur competitions: matches, tournaments, standings, and championships. Content centers on performance, tactics, and outcomes on the field. 
 • Athlete-focused updates including injuries, transfers, contracts, and retirements. Stories emphasize team dynamics and competitive impact. 
 • Coverage of leagues and events: scheduling, rule changes, drafts, and officiating controversies. The angle is sporting governance and competitive fairness. 
 • Game recaps and statistical breakdowns highlighting key plays, records, and milestones. The narrative ties individual performances to team results. 
 • Profiles and human-interest features about coaches, players, and training methods. Emphasis is on preparation, strategy, and competitive psychology. |
| **Business** 
 *Verbalizer:* "This example news text is about business news." | • Corporate news: earnings, revenue guidance, layoffs, executive changes, and strategic shifts. Articles assess company performance and shareholder impact. 
 • Markets and finance coverage: stocks, bonds, commodities, currencies, and macro sentiment. Focus is on price moves, drivers, and investor reactions. 
 • Mergers, acquisitions, IPOs, and venture funding. Pieces explain valuations, synergies, and regulatory hurdles. 
 • Industry developments such as competition, supply chains, pricing, and business models across sectors. Reporting connects firm-level actions to market structure. 
 • Policy and regulation affecting commerce: antitrust cases, trade policy, taxes, and compliance. The lens is how rules shape corporate behavior and profitability. |
| **Science & Technology** 
 *Verbalizer:* "This example news text is about science and technology." | • Scientific research findings across fields like biology, physics, medicine, and climate science. Articles emphasize methods, evidence, and potential applications or limitations. 
 • Technology product and platform news: hardware, software, mobile, cloud, and consumer gadgets. Coverage focuses on features, performance, and user impact. 
 • AI, data science, and computing breakthroughs including models, chips, algorithms, and benchmarks. Stories discuss capabilities, risks, and real-world use cases. 
 • Space and astronomy updates: launches, missions, telescopes, and planetary discoveries. Coverage highlights scientific goals and engineering challenges. 
 • Cybersecurity and privacy incidents: vulnerabilities, breaches, hacks, and defenses. Reporting centers on technical cause, affected users, and mitigations. |

Table 3: AGNews (Zhang et al., 2015a) class verbalizer and class descriptions (5 per class).

## C  EMBEDDING MODELS

Our benchmark covers ten publicly available embedding models spanning different architectures, parameter scales, and training paradigms (Table 4). We include (i) smaller sentence-transformer style encoders such as `all-MiniLM-L6-v2`, (ii) recent E5 and BGE models trained on large synthetic contrastive corpora, (iii) the GTE family including a ModernBERT-based variant, and (iv) two more recent architectures targeting embedding quality, `embeddinggemma-300m` and `Qwen3-Embedding-0.6B`. This mix allows us to test description-tuning on both lightweight models that are attractive in low-compute settings and larger, state-of-the-art encoders that already perform strongly in zero-shot retrieval and classification benchmarks.

For each model, Table 4 reports publication year, base architecture (encoder vs. decoder), backbone, main pre-training or fine-tuning corpus, parameter count, pooling strategy, and embedding dimensionality. We always use the pooling operation recommended by the model authors (e.g., mean pooling for E5, CLS pooling for GTE/BGE), and apply our description-only training on top of the publicly released checkpoints without any additional task-specific pretraining. The diversity of architectures and training recipes allows us to assess how robust our method is across model families, as presented in the main results (Table 1).

| Model | Yr | Arch. | Backbone | FT / train data | # P | Pool | Dim |
|---|---|---|---|---|---|---|---|
| `all-MiniLM-L6-v2` | 2021 | enc. | MiniLM | 1B paired sentences | 22.7M | mean | 384 |
| `e5-base-v2` | 2023 | enc. | E5 (BERT) | 270M synthetic contrastive | 110M | mean | 768 |
| `e5-large-v2` | 2023 | enc. | E5 (BERT) | same as above | 335M | mean | 1024 |
| `bge-base-en-v1.5` | 2023 | enc. | BGE (RoB.) | 1.5B pair data, contrastive | 137M | CLS | 768 |
| `bge-large-en-v1.5` | 2023 | enc. | BGE (RoB.) | same as above | 434M | CLS | 1024 |
| `gte-base-en-v1.5` | 2024 | enc.+ | GTE | MLM + contrastive pre-train | 137M | CLS | 768 |
| `gte-large-en-v1.5` | 2024 | enc.+ | GTE | same as above | 434M | CLS | 1024 |
| `gte-modernbert-base` | 2024 | enc. | ModernBERT | same as above | 149M | CLS | 768 |
| `embeddinggemma-300m` | 2025 | enc. | Gemma 3 (enc.) | Multiling. corpus (320B tok), contrastive | 308M | mean | 768[*] |
| `Qwen3-Embedding-0.6B` | 2025 | dec. | Qwen3 | synthetic multiling. contrastive | 0.6B | last | 1024 |

Table 4: Architectural and training overview of the 10 embedding models used. Columns list publication year (Yr), encoder/decoder architecture (Arch.), backbone, principal fine-tuning (FT) or pre-training data, parameter count (#P), pooling strategy (Pool), and embedding dimensionality (Dim). [*]For `embeddinggemma-300m`, dimensionality 768 corresponds to Matryoshka Representation Learning (MRL) with nested sizes 512/256/128, a training scheme enabling shorter embeddings.

## D  CONTRASTIVE LOSS VARIANTS

**Effect of the loss components.**  Table 5 compares the three objectives introduced in Section 3: the rowwise InfoNCE loss $\mathcal{L}_{\text{rows}}$, the columnwise loss $\mathcal{L}_{\text{cols}}$, and their symmetric combination $\mathcal{L} = \frac{1}{2}\mathcal{L}_{\text{rows}} + \frac{1}{2}\mathcal{L}_{\text{cols}}$. Averaged over the four datasets, the symmetric loss is always at least as good as the best of the two single-sided losses and typically improves macro-$F_1$ by a small but consistent margin. Across the ten backbones, $\mathcal{L}$ improves over $\mathcal{L}_{\text{rows}}$ by about $+0.01$ macro-$F_1$ on average and over $\mathcal{L}_{\text{cols}}$ by roughly $+0.03$ macro-$F_1$. The gains are particularly visible for the smaller or less specialized models (e.g., BGE-base, Gemma), where combining both directions yields up to $+0.04$ macro-$F_1$ compared to training only with $\mathcal{L}_{\text{cols}}$.

At the same time, the ablation confirms that $\mathcal{L}_{\text{rows}}$ is the stronger of the two components when used in isolation. For almost all backbones in Table 5, $\mathcal{L}_{\text{rows}}$ matches or slightly outperforms $\mathcal{L}_{\text{cols}}$ when trained alone, indicating that using description vectors as anchors (rows) already captures most of the benefit of description-tuning. The columnwise objective by itself is thus not sufficient to reach the best performance, but it becomes beneficial once combined with the rowwise term.

**Per-dataset behavior.**  The disaggregated results in Table 6 show that this pattern holds across tasks. Out of the 40 (model, dataset) combinations, the symmetric loss $\mathcal{L}$ achieves the best or tied-best macro-$F_1$ in 34 cases; in the remaining 6 cases, either $\mathcal{L}_{\text{rows}}$ or $\mathcal{L}_{\text{cols}}$ is better, but never by more than 0.03 absolute $F_1$. On coarse-grained topic classification (AGNews) and binary sentiment (RottenTomatoes), the three losses are often very close, with $\mathcal{L}$ typically providing a modest

refinement on top of $\mathcal{L}_{\text{rows}}$ (e.g., e5-large, gte-base). The advantage of the symmetric objective becomes more pronounced on the fine-grained tasks Banking77 and EmotionDAIR. For instance, for BGE-large on EmotionDAIR, $\mathcal{L}$ reaches 0.56 macro-$F_1$, improving over both $\mathcal{L}_{\text{rows}}$ (0.52) and $\mathcal{L}_{\text{cols}}$ (0.53); for Gemma on EmotionDAIR, the symmetric loss lifts performance from 0.45 (rows-only) and 0.54 (cols-only) to 0.57. Similar trends hold for Qwen on EmotionDAIR and for several models on Banking77.

There are a few isolated cases where a single-sided loss slightly dominates the symmetric one (e.g., Qwen on AGNews, Gemma on AGNews and RottenTomatoes, GTE-large on EmotionDAIR and RottenTomatoes), but the margins are small (1-3 $F_1$ points) and not systematic across backbones or tasks. We therefore view these as noise-level fluctuations rather than evidence against the symmetric formulation.

**Takeaways.** Overall, the ablation supports our design choice. The rowwise term $\mathcal{L}_{\text{rows}}$ is the primary driver of performance: using description embeddings as anchors already yields strong zero-shot classifiers and consistently outperforms the columnwise variant when used in isolation. The columnwise term $\mathcal{L}_{\text{cols}}$ plays a complementary role: it is weaker on its own but, when combined with $\mathcal{L}_{\text{rows}}$, acts as a regulariser that better structures the joint description-label space, leading to small but robust gains across most backbones and datasets. Consequently, we adopt the symmetric loss $\mathcal{L}$ as our default objective in all subsequent experiments.

| Model | $\mathcal{L}_{\text{rows}}$ | $\mathcal{L}_{\text{cols}}$ | $\mathcal{L}$ |
|---|---|---|---|
| **MiniLM** | | | |
| all-MiniLM-L6-v2 | 0.70 (0.20) | 0.65 (0.18) | **0.70 (0.20)** |
| **E5** | | | |
| e5-base-v2 | 0.76 (0.20) | 0.76 (0.21) | **0.77 (0.20)** |
| e5-large-v2 | 0.78 (0.19) | 0.77 (0.19) | **0.79 (0.19)** |
| **BGE** | | | |
| bge-base-en-v1.5 | 0.76 (0.20) | 0.74 (0.20) | **0.77 (0.21)** |
| bge-large-en-v1.5 | 0.78 (0.18) | 0.78 (0.18) | **0.80 (0.17)** |
| **GTE** | | | |
| gte-base-en-v1.5 | 0.77 (0.19) | 0.76 (0.19) | **0.78 (0.20)** |
| gte-modernbert-base | 0.77 (0.19) | 0.74 (0.19) | **0.77 (0.19)** |
| gte-large-en-v1.5 | 0.77 (0.19) | 0.77 (0.19) | **0.78 (0.19)** |
| **Qwen** | | | |
| Qwen3-Embedding-0.6B | 0.80 (0.16) | 0.76 (0.18) | **0.80 (0.16)** |
| **Gemma** | | | |
| embeddinggemma-300m | 0.71 (0.19) | 0.72 (0.16) | **0.75 (0.15)** |

Table 5: Mean macro-F1 (std. in parentheses) for each loss, averaged over the four datasets AGNews, Banking77, EmotionDAIR and RottenTomatoes. Best loss per model (row) is in bold.

| Model | AGNews | | | Banking77 | | | EmotionDAIR | | | RottenTomatoes | | |
|---|---|---|---|---|---|---|---|---|---|---|---|---|
| | $\mathcal{L}_{\text{rows}}$ | $\mathcal{L}_{\text{cols}}$ | $\mathcal{L}$ | $\mathcal{L}_{\text{rows}}$ | $\mathcal{L}_{\text{cols}}$ | $\mathcal{L}$ | $\mathcal{L}_{\text{rows}}$ | $\mathcal{L}_{\text{cols}}$ | $\mathcal{L}$ | $\mathcal{L}_{\text{rows}}$ | $\mathcal{L}_{\text{cols}}$ | $\mathcal{L}$ |
| **MiniLM** | | | | | | | | | | | | |
| all-MiniLM-L6-v2 | 0.78 | 0.71 | **0.79** | 0.89 | 0.80 | **0.90** | 0.43 | 0.38 | **0.43** | 0.69 | 0.70 | **0.70** |
| **E5** | | | | | | | | | | | | |
| e5-base-v2 | 0.78 | 0.80 | **0.81** | 0.95 | 0.95 | **0.96** | 0.48 | 0.45 | **0.48** | 0.82 | **0.83** | 0.82 |
| e5-large-v2 | 0.81 | 0.80 | **0.82** | 0.95 | 0.95 | **0.96** | 0.52 | 0.51 | **0.53** | 0.85 | 0.84 | **0.86** |
| **BGE** | | | | | | | | | | | | |
| bge-base-en-v1.5 | 0.81 | 0.75 | **0.82** | 0.95 | 0.94 | **0.95** | 0.47 | 0.47 | **0.47** | 0.81 | 0.81 | **0.82** |
| bge-large-en-v1.5 | 0.82 | 0.80 | **0.82** | 0.94 | 0.94 | **0.95** | 0.52 | 0.53 | **0.56** | 0.84 | 0.84 | **0.85** |
| **GTE** | | | | | | | | | | | | |
| gte-base-en-v1.5 | 0.82 | 0.80 | **0.83** | 0.94 | 0.92 | **0.95** | 0.49 | 0.48 | **0.49** | 0.83 | 0.84 | **0.85** |
| gte-modernbert-base | **0.80** | 0.71 | 0.80 | 0.94 | 0.92 | **0.94** | 0.49 | 0.48 | **0.49** | 0.83 | 0.84 | **0.84** |
| gte-large-en-v1.5 | 0.82 | 0.79 | **0.83** | 0.93 | 0.94 | **0.95** | 0.50 | **0.51** | 0.50 | 0.84 | **0.86** | 0.83 |
| **Qwen** | | | | | | | | | | | | |
| Qwen3-Embedding-0.6B | **0.86** | 0.78 | 0.85 | 0.90 | 0.90 | **0.92** | 0.56 | 0.50 | **0.57** | 0.87 | 0.87 | **0.88** |
| **Gemma** | | | | | | | | | | | | |
| embeddinggemma-300m | **0.75** | 0.71 | 0.74 | 0.91 | 0.92 | **0.94** | 0.45 | 0.54 | **0.57** | 0.73 | **0.74** | 0.73 |

Table 6: Disaggregated macro-$F_1$ by dataset and loss. Entries are $F_1$ scores; for each model and dataset, the best loss is in bold.

# E ROBUSTNESS TO NOISY DESCRIPTIONS.

To test how sensitive our method is to imperfect descriptions, we run a controlled corruption experiment on AGNews using `embeddinggemma-300m`. Starting from the original description set, we progressively replace a fraction of descriptions with overly vague, non-discriminative sentences (examples in Table 8). We vary the noise level in (0.0, 0.25, 0.5, 1.0), where (1.0) means that *all* descriptions are vague. For each noise level and each loss ($\mathcal{L}_{\text{rows}}$, $\mathcal{L}_{\text{cols}}$, and the symmetric $\mathcal{L}$), we train with the learning rate selected by the uniformity heuristic and track macro-$F_1$ over training (Figure 4). Table 7 reports the final $F_1$ scores (taking into account early stopping).

| Noise level | $\mathcal{L}_{\text{rows}}$ | $\mathcal{L}_{\text{cols}}$ | $\mathcal{L}$ |
|---|---|---|---|
| 0.00 | 0.75 | 0.71 | 0.74 |
| 0.25 | 0.64 | **0.71** | 0.67 |
| 0.50 | 0.59 | **0.64** | 0.61 |
| 1.00 | 0.41 | **0.45** | 0.35 |

Table 7: Macro-$F_1$ on AGNews for `embeddinggemma-300m` under different description noise levels and losses.

With clean descriptions ((0.0) noise), the rowwise loss is strongest (0.75 vs. 0.71 for $\mathcal{L}_{\text{cols}}$), and the symmetric loss ($\mathcal{L}$) is only slightly behind at 0.74. Once we inject moderate noise (0.25 and 0.5), the behavior changes markedly. $\mathcal{L}_{\text{rows}}$ degrades sharply (down to 0.64 and 0.59, i.e. $-0.11$ and $-0.16$ absolute), whereas $\mathcal{L}_{\text{cols}}$ is much more stable: at 25% noise it remains essentially unchanged compared to the clean setting, and at 50% noise it still clearly dominates the other objectives (0.64 vs. 0.59 and 0.61). The symmetric objective $\mathcal{L}$ interpolates between these regimes: with clean descriptions it tracks $\mathcal{L}_{\text{rows}}$, but under moderate noise it moves closer to $\mathcal{L}_{\text{cols}}$, preserving much of the robustness provided by the column term. At the extreme noise level (1.0), when all descriptions are vague, all three objectives degrade substantially, as expected when the supervision signal is entirely uninformative.

These trends align with the analytical properties of $\mathcal{L}_{\text{cols}}$. By using a log-sum-exp over multiple positives per label, it (i) reweights descriptions so that strong, representative ones dominate the gradient while noisy/vague ones are down-weighted, and (ii) adds extra smoothing / regularization

at the label level. In the moderate-noise regime, arguably the most realistic setting, this makes $\mathcal{L}_{\text{cols}}$ significantly more robust than a pure rowwise InfoNCE. The symmetric loss $\mathcal{L}$ largely inherits this robustness while retaining strong performance in the clean setting, indicating that combining row- and columnwise objectives yields a good trade-off between peak accuracy and resilience to imperfect description sets.

| # | Vague AGNews description |
|---|---|
| 1 | Coverage about the news, containing information about events and happenings in the world. The text discusses topics that may be of interest to readers who follow current events. |
| 2 | A piece of text about something that was reported by journalists or news organizations. It contains sentences and paragraphs describing various matters. |
| 3 | This is a news article that provides information to readers. The content covers subjects that are deemed newsworthy by editors and reporters. |
| 4 | Information presented in written form about things that have occurred or are occurring. The article is structured with a headline and body text. |
| 5 | Some content about a topic that readers might find relevant. The writing style follows journalistic conventions and presents facts or opinions. |

Table 8: Examples of deliberately vague AGNews descriptions used to inject noise into the label description set.

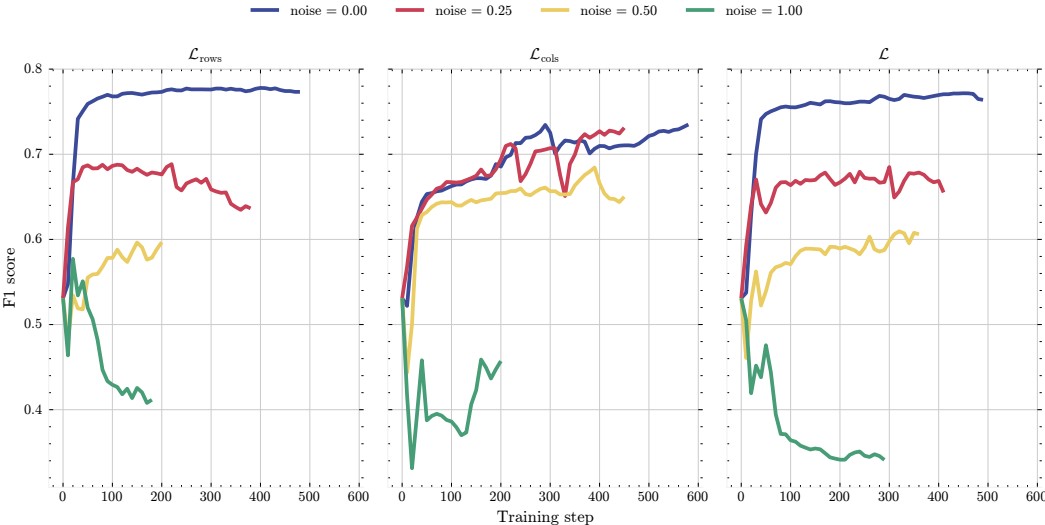

Figure 4: Macro-$F_1$ on AGNews for `embeddinggemma-300m` under increasing description noise. Each panel shows training curves for one loss ($\mathcal{L}_{\text{rows}}$, $\mathcal{L}_{\text{cols}}$, and the symmetric $\mathcal{L}$); colors denote noise levels (0.0, 0.25, 0.5, 1.0). The columnwise loss remains markedly more stable under moderate noise, and the symmetric loss largely inherits this robustness.

## F    ROBUSTNESS TO IMBALANCED DESCRIPTION COUNTS

Figure 5 studies whether the proposed method is sensitive to imbalances in the number of descriptions per label. We use `all-MiniLM-L6-v2` on AGNews and compare three setups: (i) the base model without description training, (ii) a balanced configuration with the same number of descriptions per label, and (iii) an imbalanced configuration where *Business* has 10 descriptions while the remaining labels have only 3. For the symmetric objective $\mathcal{L}$ (left panel), adding descriptions improves $F_1$ across all labels relative to the base model, and giving extra descriptions to *Business* yields a small additional gain for that label while *Science/Tech*, *Sports*, and *World* remain essentially unchanged between the balanced and imbalanced setups. In contrast, under the pure rowwise loss $\mathcal{L}_{\text{rows}}$ (right panel), *Business* again benefits from the extra descriptions, but the other three labels incur a small $F_1$ drop when moving from the balanced to the imbalanced configuration (on the order of 1-2

points), although they still clearly outperform the base model. This pattern is consistent with the role of the columnwise term in $\mathcal{L}$. Because $\mathcal{L}_{\text{cols}}$ optimizes a per-label normalized quantity $Z_\ell^+/Z_\ell$, it is inherently less sensitive to how many descriptions a label has: each label's objective is scaled by its own normalizer rather than competing directly with other labels for probability mass. As a result, the symmetric loss $\mathcal{L}$, which includes $\mathcal{L}_{\text{cols}}$, can absorb an over-representation of *Business* without harming the other classes, whereas the purely rowwise objective shows mild cross-label interference when one label receives substantially more descriptions than the rest.

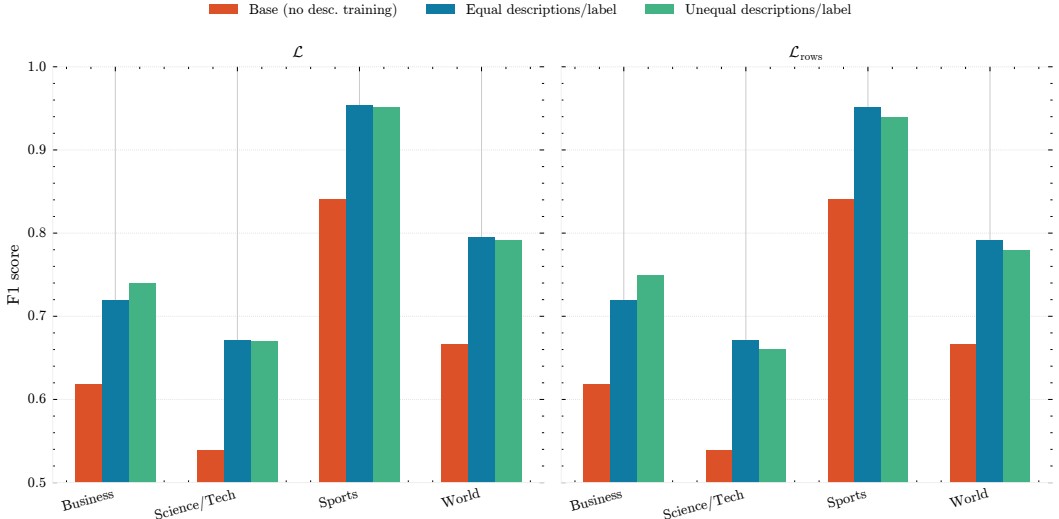

Figure 5: Per-label $F_1$ on AGNews for `all-MiniLM-L6-v2` under different description allocation schemes. We compare the base model (no description training), a balanced setup with an equal number of descriptions per label, and an imbalanced setup where *Business* has 10 descriptions while the other labels have only 3. Left: symmetric loss $\mathcal{L}$, where extra descriptions for *Business* slightly improve that label without affecting the others. Right: rowwise loss $\mathcal{L}_{\text{rows}}$, where *Business* gains while the remaining labels incur a small $F_1$ drop but still outperform the base model.

## G    EFFECT OF THE NUMBER OF DESCRIPTIONS PER LABEL

Figure 6 investigates how performance varies with the number of descriptions per label when using the default symmetric loss $\mathcal{L}$. We vary $K \in \{1, 3, 5, 10, 20\}$ while keeping verbalizers fixed; rows correspond to `e5-base-v2` and `e5-large-v2`, columns to the four datasets. Across all model-dataset pairs, using a single description per label consistently underperforms, while moving from 1 to 3-5 descriptions yields a substantial jump in macro-$F_1$. Beyond $K = 5$, the curves flatten and often become mildly non-monotonic: additional descriptions bring at most small gains (e.g. EmotionDAIR and RottenTomatoes) and sometimes slight degradation (e.g. AGNews and Banking77).

We believe this pattern is largely driven by *description quality* rather than by the number of descriptions per se. In practice, we curated descriptions in a sensible order, using the clearest, most representative ones first. As $K$ grows, it becomes increasingly difficult to add new descriptions that are both (i) genuinely novel and (ii) still precise and label-specific; later descriptions tend to be either redundant or more generic. From the model's perspective, increasing $K$ therefore shifts the label description set from "a few high-quality positives" to "a mixture of strong and weaker positives," effectively introducing a mild form of noise. This interpretation is consistent with our explicit noise ablation in Section E: the method is robust to moderate levels of noisy descriptions, but performance plateaus or slightly declines once weaker descriptions start to dominate the marginal additions. Overall, the curves suggest that 3-5 carefully written descriptions per label capture most of the benefit of description-tuning, and that pushing to 10-20 descriptions is only worthwhile if one can maintain comparable quality.

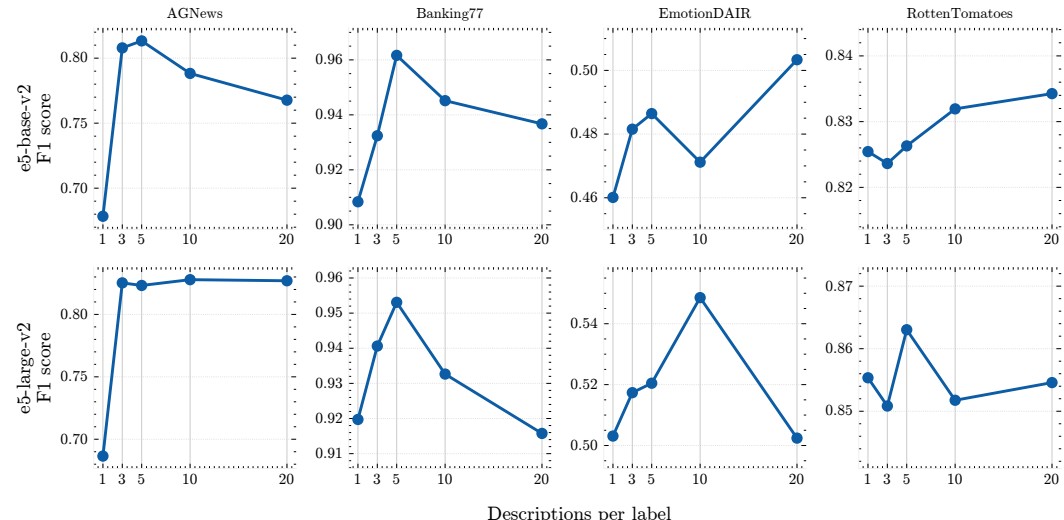

Figure 6: Effect of the number of descriptions per label on macro-$F_1$ for the symmetric loss $\mathcal{L}$. Rows correspond to `e5-base-v2` (top) and `e5-large-v2` (bottom); columns correspond to AGNews, Banking77, EmotionDAIR, and RottenTomatoes. Each curve plots macro-$F_1$ as the number of descriptions per label $K$ increases from 1 to 20. Performance improves sharply when going from 1 to 3-5 descriptions and then flattens or becomes slightly non-monotonic as additional descriptions are added, which we attribute to the increasing difficulty of generating novel yet high-quality descriptions at larger $K$.

## H    CHOICE OF INFERENCE ANCHOR

We finally ablate the choice of label representation ("anchor") used during training and inference. We compare three modes:

1. **Label.** We replace the verbalizer sentence by the raw label string (e.g., "Sports") and use this as the anchor both during training and at inference. 2. **Verbalizer.** Our default setting: we train and evaluate with the full verbalizer sentence (e.g., "This news snippet is about sports."). 3. **Mean.** We train as in the default setting (with verbalizers), but at inference we instead score documents against the mean embedding of each label's descriptions, $\frac{1}{K_y} \sum_k e(d_y^k)$.

Figure 7 reports macro-$F_1$ for `e5-base-v2` and `e5-large-v2` across the four datasets. No single mode dominates everywhere. For AGNews, the verbalizer gives the best performance; for Banking77 and EmotionDAIR, using the raw label string tends to outperform the verbalizer; and for RottenTomatoes the mean-of-descriptions is strongest. Across all model-dataset pairs, however, the differences are small (typically within 1-2 $F_1$ points), and we do not observe statistically significant differences between the three modes. This indicates that the method is not fragile to the precise choice of anchor. Taken together, these results suggest that (i) training directly on label names is a viable alternative to full verbalizers, and (ii) even when training with verbalizers, practitioners can safely switch to mean description embeddings at inference time, which typically match the performance of the default.

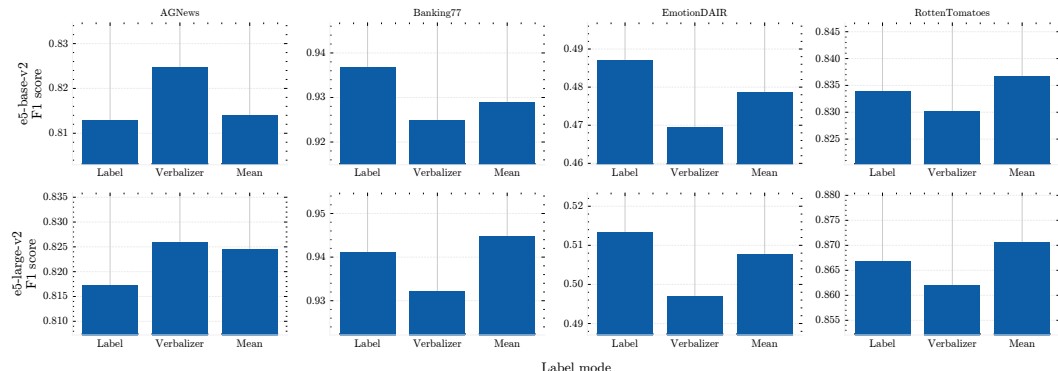

Figure 7: Ablation over label representation. Bars show macro-$F_1$ when training and evaluating on the raw label string (*Label*), training and evaluating on the full verbalizer sentence (*Verbalizer*, our default), or training on verbalizers but using the mean description embedding at inference (*Mean*). Rows correspond to `e5-base-v2` (top) and `e5-large-v2` (bottom); columns correspond to AG-News, Banking77, EmotionDAIR, and RottenTomatoes. The best choice varies mildly by dataset, but the small differences are not statistically significant, indicating robustness to the particular inference anchor.

## I   PERFORMANCE ON LARGE LABEL SPACES

Finally, we test whether description-tuning remains effective when the label space is large. We use the full 77-way *Banking77* label set, writing only three descriptions per label, and evaluate on the official test split. For the final runs, we train `all-MiniLM-L6-v2` with a learning rate of $1 \times 10^{-5}$ and `embeddinggemma-300m` with $1 \times 10^{-4}$, both selected via the uniformity-based procedure from Section 3.

Table 9 shows that even under this harder setting, description-tuning yields substantial gains. For `all-MiniLM-L6-v2`, macro-$F_1$ and accuracy increase by $+0.07$ ($0.59 \rightarrow 0.66$), while precision and recall both improve, with recall rising more strongly ($0.62 \rightarrow 0.69$). The effect is even more pronounced for `embeddinggemma-300m`: macro-$F_1$ improves by $+0.09$ ($0.52 \rightarrow 0.61$) and accuracy by $+0.08$, with recall jumping from 0.55 to 0.64 ($+0.10$) and precision from 0.60 to 0.64 ($+0.03$). In other words, the method substantially reduces false negatives, crucial in a large label space, without sacrificing precision.

Overall, these results indicate that our description-only finetuning scales to high-cardinality label spaces even with a very small description budget (three descriptions per label). The gains in macro-$F_1$ and recall suggest that aligning labels to a compact set of descriptions helps the model better cover many fine-grained intents rather than merely sharpening predictions for the most frequent ones.

| Model | $F_1$ | Acc. | Prec. | Rec. |
|---|---|---|---|---|
| all-MiniLM-L6-v2 | 0.59 | 0.59 | 0.65 | 0.62 |
| trained | 0.66 (+0.07) | 0.66 (+0.07) | 0.69 (+0.04) | 0.69 (+0.07) |
| embeddinggemma-300m | 0.52 | 0.52 | 0.60 | 0.55 |
| trained | 0.61 (+0.09) | 0.60 (+0.09) | 0.64 (+0.03) | 0.64 (+0.10) |

Table 9: Base vs. trained performance on Banking77. Trained rows show absolute improvements in parentheses.

## J  LEARNING-RATE SELECTION VIA UNIFORMITY

**Uniformity-based learning rate selection.**  As discussed in Section 3, we treat the learning rate (LR) as the main sensitive hyperparameter and select it using a *label-free* criterion based on the uniformity loss $\mathcal{L}_{\text{uni}}$ in equation 4. For each model-dataset pair, we run short warmup trainings over a small grid of candidate LRs and evaluate $\mathcal{L}_{\text{uni}}$ on an unlabeled pool $\mathcal{X}_u$ from the target domain. We then choose the LR that minimizes $\mathcal{L}_{\text{uni}}$, i.e., that yields the most uniform (least collapsed) document embeddings on $\mathcal{X}_u$. Table 10 lists the selected LRs; these are the hyperparameters used for all results reported in the main table (Table 1).

| Model | AGNews | Banking77 | EmotionDAIR | RottenTomatoes |
|---|---|---|---|---|
| all-MiniLM-L6-v2 | $1 \cdot 10^{-5}$ | $1 \cdot 10^{-4}$ | $1 \cdot 10^{-4}$ | $3 \cdot 10^{-5}$ |
| e5-base-v2 | $1 \cdot 10^{-4}$ | $3 \cdot 10^{-4}$ | $3 \cdot 10^{-4}$ | $1 \cdot 10^{-4}$ |
| e5-large-v2 | $1 \cdot 10^{-4}$ | $1 \cdot 10^{-4}$ | $1 \cdot 10^{-4}$ | $5 \cdot 10^{-5}$ |
| bge-base-en-v1.5 | $5 \cdot 10^{-4}$ | $3 \cdot 10^{-4}$ | $5 \cdot 10^{-4}$ | $3 \cdot 10^{-5}$ |
| bge-large-en-v1.5 | $1 \cdot 10^{-4}$ | $1 \cdot 10^{-4}$ | $3 \cdot 10^{-4}$ | $5 \cdot 10^{-6}$ |
| gte-base-en-v1.5 | $1 \cdot 10^{-4}$ | $1 \cdot 10^{-4}$ | $5 \cdot 10^{-4}$ | $1 \cdot 10^{-5}$ |
| gte-modernbert-base | $5 \cdot 10^{-4}$ | $5 \cdot 10^{-4}$ | $5 \cdot 10^{-4}$ | $5 \cdot 10^{-4}$ |
| gte-large-en-v1.5 | $3 \cdot 10^{-4}$ | $5 \cdot 10^{-5}$ | $3 \cdot 10^{-4}$ | $1 \cdot 10^{-4}$ |
| Qwen3-Embedding-0.6B | $3 \cdot 10^{-5}$ | $1 \cdot 10^{-5}$ | $3 \cdot 10^{-5}$ | $1 \cdot 10^{-5}$ |
| embeddinggemma-300m | $1 \cdot 10^{-4}$ | $3 \cdot 10^{-5}$ | $5 \cdot 10^{-5}$ | $5 \cdot 10^{-5}$ |

Table 10: Learning rates selected by minimizing the uniformity loss ($\mathcal{L}_{\text{uni}}$) on an unlabeled pool ($\mathcal{X}_u$) for each model–dataset pair. The LRs are used in the main results (Table 1).

**Relationship between uniformity and downstream performance.**  Appendix Figure 8 examines how $\mathcal{L}_{\text{uni}}$, measured on $\mathcal{X}_u$, relates to downstream macro-$F_1$ after description-only training. Our goal is not to treat $\mathcal{L}_{\text{uni}}$ as a perfect surrogate for macro-$F_1$, but as a practical label-free *heuristic* for LR selection. Across the 40 model-dataset pairs in Figure 8, **29/40** exhibit a statistically significant *negative* Pearson correlation between $\mathcal{L}_{\text{uni}}$ and macro-$F_1$ ($p \leq 0.10$): runs that yield more uniform embeddings on $\mathcal{X}_u$ (lower $\mathcal{L}_{\text{uni}}$) tend to achieve higher macro-$F_1$. The remaining **11/40** pairs show weak, non-significant correlations, which are typically slightly negative or close to zero but never strongly positive. In other words, lower uniformity does not guarantee higher macro-$F_1$, yet it is empirically helpful in the majority of cases and, critically, does not appear to systematically harm performance in the remainder. This supports using $\mathcal{L}_{\text{uni}}$ as a robust, label-free signal for picking a reasonable learning rate in true zero-shot settings, while acknowledging that it is empirically useful but not universally predictive.

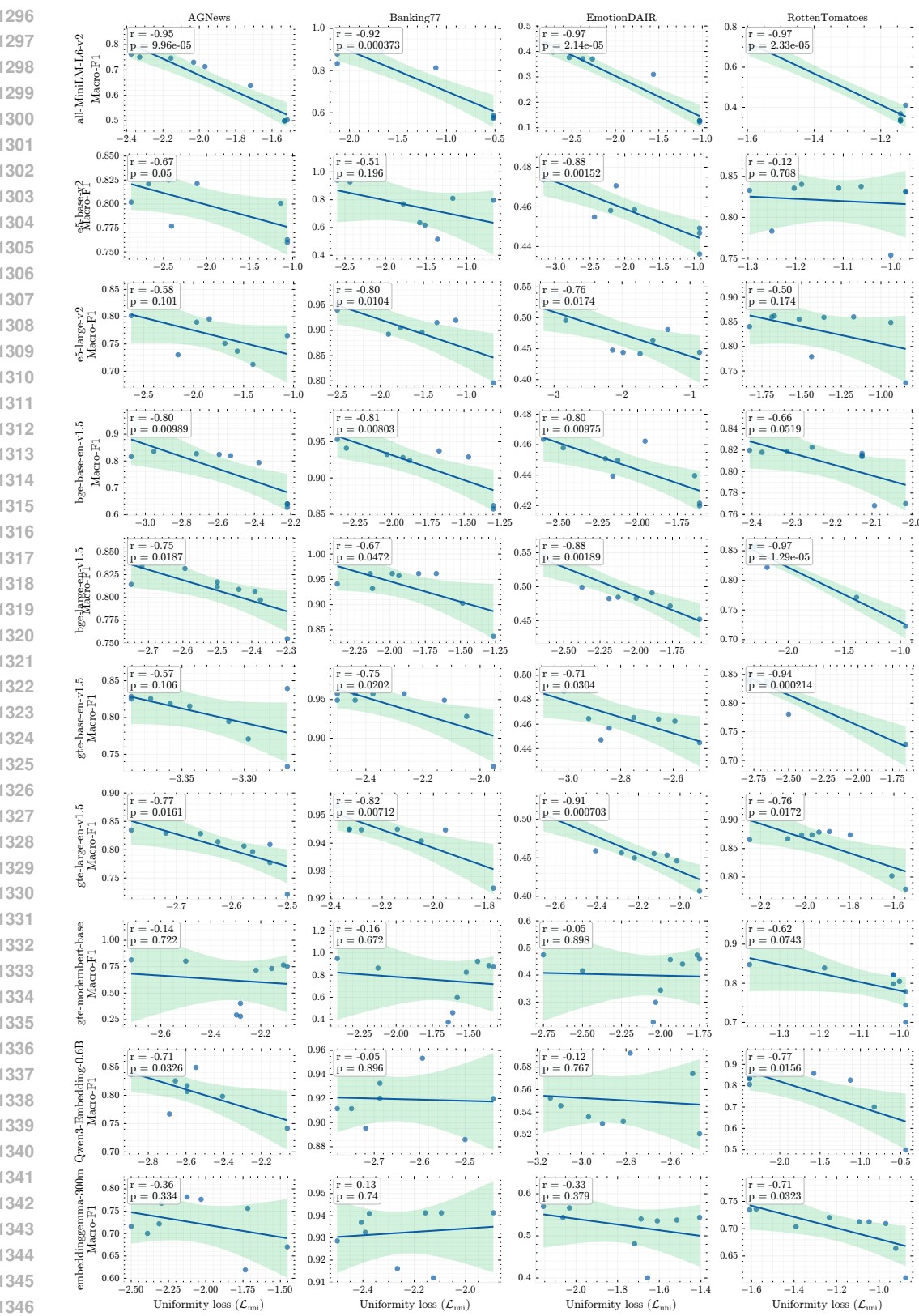

Figure 8: Scatter plots of uniformity loss $\mathcal{L}_{\mathrm{uni}}$ against Macro-F1 performance across datasets. Rows correspond to embedding models, while columns correspond to datasets. Each subplot shows individual runs with a different learning rate (dots), an ordinary least squares regression line with 95% confidence interval (shaded), and the Pearson correlation coefficient between $\mathcal{L}_{\mathrm{uni}}$ and Macro-F1.

