# OpenReview forum: "Description-Only Supervision: Contrastive Label–Embedding Alignment for Zero-Shot Text Classification"
_ICLR.cc/2026/Conference — Submitted to ICLR 2026_

### Official Review · Reviewer_gUCW · 2025-10-27

**Soundness:** 1
**Presentation:** 1
**Contribution:** 1
**Rating:** 2
**Confidence:** 2

**Summary:**

This paper presents a contrastive learning framework for zero-shot text classification using only label verbalizers and a small set of natural language descriptions per label. The method applies a dual InfoNCE objective to align verbalizers and descriptions, claiming to improve performance without using any labeled documents. Experiments across multiple datasets and embedding models show consistent macro-F1 gains over zero-shot baselines, and improved stability compared to few-shot SetFit.

**Strengths:**

* Conceptually simple and lightweight method, requiring no labeled training documents.

* Empirical results are broad and convincing, covering 10 encoders and 4 datasets.

* Demonstrates consistent performance improvements and low variance across random runs.

**Weaknesses:**

* Novelty concern:
The idea of using natural language descriptions for label supervision in zero or few-shot classification has been explored in prior work. This paper applies a contrastive objective to align label verbalizers with their descriptions, but this formulation may not represent a substantial conceptual advance. The core component, which are label descriptions, contrastive learning, and dual encoders, are already widely used.
Furthermore, it is not clear how description-only supervision fundamentally differs from conventional label supervision. Although the authors emphasize that no labeled documents are used, the descriptions are manually written for each label, and thus, still reflect explicit labeling information. In this sense, the line between using label supervision and using label descriptions remains blurred and requires clearer theoretical or empirical justification.

* No ablation studies: The contribution of each component (e.g., dual InfoNCE, verbalizer vs. description) is not isolated. How much gain comes from descriptions alone versus contrastive finetuning?

* Poorly annotated appendix:
Tables and figures in the appendix are presented without explanatory text, reducing clarity and reproducibility.

**Questions:**

* Novelty and supervision scope:
The proposed method builds on well-known components (verbalizers, natural language label descriptions, and contrastive objectives) all of which have been explored in prior work on zero or few-shot classification. While the paper emphasizes that no labeled documents are used, the manually authored descriptions per label still constitute explicit supervision.
What exactly distinguishes this setup from conventional label supervision?
And what is the substantive novelty beyond recombining existing elements in a contrastive learning framework? A clearer conceptual delineation is needed to justify the claimed contribution.

**Details Of Ethics Concerns:**

None.

---

> ### Author Response · Authors · 2025-11-20
> **Author response to Reviewer gUCW – initial reply**
>
> **Novelty and supervision scope**
>
>
> Thank you for the detailed comments on novelty and supervision. We agree that the individual components we rely on, natural language descriptions, contrastive learning, and dual encoders, are well established. Our aim is not to present these elements as new, but to show that combining them in a specific way addresses a practical gap: improving zero-shot text classification without using any labeled documents, while retaining dual-encoder efficiency.
>
> The conceptual contribution lies in *how* these ingredients are combined. We shift the supervision signal entirely off documents and onto the **label–description relation**, and use this to re-shape a dual-encoder embedding space through a **symmetric, multi-positive contrastive objective**, all while preserving standard dual-encoder zero-shot inference.
>
> Concretely, compared to prior description- or entailment-based ZSC work:
>
> * **Most description/NLI methods** use a **cross-encoder** or prompting setup that jointly encodes document + label (or label definition) at inference time. This:
>
>   * makes inference cost scale with the number of labels,
>   * prevents pre-computation of embeddings, and
>   * typically does *not* explicitly model the many-to-one mapping between multiple descriptions and a single label.
>
> * Methods that keep **dual-encoder inference** often either:
>
>   * fine-tune on labeled document–label pairs, or
>   * only plug descriptions in at inference (e.g., by comparing documents to label texts) without adapting the encoder using description-only supervision.
>
> In contrast, our method:
>
> 1. **Preserves dual-encoder inference.**
>    We keep the deployment regime of a dual encoder: labels (verbalizers) and documents are encoded independently, can be cached, and classification at test time is simple nearest-neighbor similarity between document embeddings and label verbalizers. There is no cross-encoder or document–definition scoring at inference.
>
> 2. **Uses a description-only, multi-positive contrastive objective.**
>    Training uses *only* label verbalizers and a few free-form natural-language descriptions per label; no labeled documents. The **column-wise multi-positive InfoNCE term** explicitly treats each label as having multiple positive descriptions, with a log-sum-exp aggregation that:
>
>    * captures the many-to-one label–description relationship,
>    * provides adaptive weighting over positives (down-weighting outlier descriptions), and
>    * directly shapes the label embedding geometry.
>
>   This allows to move the supervision signal entirely off documents and onto the label–description relation itself, and uses that to re-shape a dual-encoder embedding space.
>
> 3. **Introduces symmetric row–column alignment.**
>    We combine row-wise (description→verbalizer) and column-wise (verbalizer→descriptions) objectives, creating bidirectional alignment forces that pull each verbalizer toward the “high-density region” of its description cloud while repelling it from other labels. This dual-anchor formulation is specifically designed for the label–description structure rather than generic instance-instance contrastive learning.
>
> In summary, prior description/entailment ZSC approaches either sacrifice computational efficiency (cross-encoders) or use descriptions in a more ad-hoc way at inference. Our contribution is to show how to **use descriptions as the sole supervision signal to learn a label-specific embedding geometry for an off-the-shelf dual encoder, via a symmetric multi-positive contrastive objective, while retaining efficient, pre-encodable dual-encoder inference at test time.**
>
>
>
> **Missing ablations**
>
> We fully agree that ablations are necessary to disentangle the contributions of each component. As noted in our response to Reviewer hnXM, we will include:
>
> * Ablations of the loss components ($L_{\text{rows}}$ only, $L_{\text{cols}}$ only, full symmetric loss),
> * Ablations of the inference anchor (verbalizer vs mean of descriptions), and
> * Ablations comparing full verbalizer sentences vs simple label words.
>
> These will be reported on a representative subset of models and datasets, and will help clarify how much gain comes from descriptions alone vs description-only finetuning, and which design choices are most important.
>
>
> **Poorly annotated appendix**
>
> We appreciate this criticism. The current appendix indeed contains tables and figures with minimal explanatory text. In the revision, we will:
>
> * Add descriptive captions and short surrounding paragraphs explaining what each table/figure shows and how to interpret it,
> * Cross-reference main-text claims to the corresponding appendix sections, and
> * Ensure that the appendix is self-contained enough to support reproducibility.
>
>
> If there are specific issues among these that you view as decisive for your final recommendation, or additional analyses you consider crucial, we would appreciate a brief indication so we can prioritize them in the revision.

---

> > ### Comment · Reviewer_gUCW · 2025-11-25
> >
> > Thank you for the clarifications. However, I cannot reassess the contribution without seeing the actual experimental results. Once the revised submission includes the ablations and comparisons you described, I will be able to make a more informed judgment and reconsider my score.

---

> > > ### Author Response · Authors · 2025-12-04
> > > **Final Response to gUCW**
> > >
> > > Thank you for your follow-up and for being willing to reconsider your assessment once additional results were available.
> > >
> > > In the revised manuscript we have now incorporated the ablations and comparisons outlined in our initial response, including:
> > >
> > > * Loss-component ablations for L_rows, L_cols and their symmetric combination, as well as anchor and verbalizer/label-word variants (Appendix D–F–H), to disentangle the contribution of each design choice and of description-only fine-tuning itself.
> > > * A large-label experiment on the full 77-class Banking77 task together with a discussion of the (O(D\cdot L)) similarity matrix in terms of memory and latency (Appendix I).
> > > * A clearer positioning of the conceptual contribution relative to prior description-/entailment-based ZSC and dual-encoder methods in the Introduction and Related Work.
> > > * Substantially expanded and better annotated appendix sections (Appendix A–J), with descriptive captions and explanatory text so that each set of tables and figures can be read and interpreted independently.
> > >
> > > We hope these additions address your concerns about novelty, supervision scope, and missing ablations, and that the revised version provides a clearer and more complete basis for your final evaluation.

---

### Official Review · Reviewer_87iw · 2025-10-30

**Soundness:** 2
**Presentation:** 1
**Contribution:** 2
**Rating:** 2
**Confidence:** 3

**Summary:**

This paper proposes description-only supervision for zero-shot text classification by aligning label verbalizers with small sets of human-written label descriptions using a dual-direction contrastive loss (row-wise and column-wise InfoNCE). Experiments on four common classification datasets with ten embedding models show improvements over naive zero-shot baselines, while claiming lower supervision costs than few-shot alternatives.

**Strengths:**

1. Uses only label descriptions, preserving dual-encoder efficiency (cacheable embeddings; label scaling). Clear deployment upside vs. cross-encoders/ICL. Clean, principled objective: The row-wise InfoNCE + column-wise multi-positive formulation captures one-to-many label–description relationships and adaptively reweights positives. The derivation and gradients are explicit.

2.  +0.10 macro-F1 on average across 10 encoders × 4 tasks; especially large relative lifts for small models. Comprehensive per-family analysis.

3. The uniformity criterion is a neat, inexpensive heuristic to prevent collapse in small-data contrastive tuning.

4. Early stopping, small description sets, and minimal engineering make replication/deployment feasible.

5. The visualized results are visually appealing.

**Weaknesses:**

1. Banking77 is restricted to six card-related intents, which may understate difficulty relative to the full 77-class benchmark; generality to large label spaces remains partially untested in this paper’s main results.

2. The uniformity-based LR selection samples pairs from the test subset of the target domain, which can blur the line between tuning and evaluation (even though labels are not used). A cleaner protocol (dev split) would avoid potential leakage.

3. Authors fix five generic descriptions per class and postpone quality optimization; robustness to noisy/misaligned descriptions is not systematically ablated.

4. While SetFit is a fair few-shot comparison, contemporary description-driven ZSC methods (e.g., NLI-style label entailment or richer definition-based approaches) aren’t exhaustively compared under identical dual-encoder constraints.

5. The paper states that Figure 1 demonstrates the core idea (Line 131), yet the figure mainly illustrates UMAP embeddings for AGNews rather than providing a conceptual or architectural depiction of the proposed framework.

6. Poor writing, formatting, and referencing quality.   Inconsistent formula numbering: some equations are labeled, while others are not.   Reference formatting is not standardized, and several citations contain extra parentheses (e.g., Lines 45, 50, 64).  Overall layout lacks polish.

**Questions:**

1. You compute uniformity on pairs sampled from the test subset (labels unused). Could you report results using a separate validation split for uniformity selection to rule out any subtle overfitting and quantify the gap (if any)?
2. How does performance scale from 1→3→5→10 descriptions per label, and how sensitive is the method to noisy or partially off-topic descriptions? An ablation would help practitioners budget description effort.
3. Have you tried full Banking77 (77 classes) or other datasets with dozens–hundreds of labels? How does the O(DL) batch construction behave in memory/latency, and does the column-wise term remain stable when K varies widely across labels?
4. Many zero-shot methods compare documents to labels via entailment or rich label definitions. Could you include a dual-encoderized NLI/definition baseline (not cross-encoder) to isolate the value of the proposed contrastive training?
5. You argue lower uniformity correlates with better F1. Can you provide per-dataset correlation plots across more models or show cases where the correlation breaks, to bound the reliability of this selection rule? (Some plots are in the appendix; more discussion would help.)
6. Since descriptions are lightweight to write, have you evaluated cross-domain transfer (e.g., train verbalizer/description alignment on one domain and test on another) or multilingual zero-shot where label descriptions are translated?
7. What clear conceptual advancement distinguishes this method from prior description- or entailment-based ZSC work?

---

> ### Author Response · Authors · 2025-11-20
> **Author response to Reviewer 87iw – initial reply (I/II)**
>
> **Restricted Banking77 subset and large label spaces**
>
> Our initial motivation for restricting Banking77 to six card-related intents was to focus on a more fine-grained subset where subtle semantic distinctions matter. We agree that this choice understates the challenge relative to the full 77-class benchmark and that evaluating on larger label spaces is important to assess generality and the behavior of the column-wise term.
>
> In the revision, we will:
>
> * Add at least one **large-label experiment**, and
> * Report both performance and the memory/latency behavior of the $O(D \cdot L)$ batch construction.
>
> This will directly address the concern about generality to larger label spaces.
>
> **Uniformity-based LR selection and dev split**
>
> You are right that in the current setup we compute uniformity on pairs sampled from the **test subset**, without labels. This was intentional: it reflects a realistic ZSC scenario where a practitioner has access to a *single* pool of unlabeled documents from the target domain and uses a random subset of that pool to select the LR, then runs inference on the full set.
>
> We understand the concern about separating the data used for LR selection from the data used for reporting metrics. During initial experimentation, we also tried a variant with a dedicated **unlabeled dev split**, and the results appeared stable. In the revision, we will:
>
> * Add an LR-selection variant that uses a disjoint unlabeled dev subset;
> * Report the corresponding macro-F1 alongside the original protocol; and
> * Quantify any differences to show whether this choice has any practical impact.
>
> Labels are never used for LR selection; the only change is which unlabeled subset is used to compute uniformity.
>
> **Description robustness (number and noise)**
>
> We agree this is important and fully aligns with your suggestion. As noted in our response to Reviewer hnXM, we will add ablations where we vary the number of descriptions per label (1→3→5→10) and inject noise into a subset of descriptions. This will help practitioners decide how much effort to spend on descriptions and how tolerant the method is to imperfect label metadata.
>
> **Comparison to description-driven baselines / dual-encoderized NLI**
>
> Thank you for pointing out the need to make this baseline clearer. Our current **zero-shot baseline** is indeed a dual-encoder setup with labels verbalized as semantically rich sentences and classification performed by comparing document embeddings to these label representations via cosine similarity (i.e., a “definition / description” baseline under dual-encoder constraints). Table 1 reports results for this baseline under the original model name and for our method as the "trained" variation. We will clarify this more explicitly in the paper to avoid confusing.
>
> **Figure 1 vs conceptual depiction**
>
> You are correct that Figure 1, as currently presented, is more of a *UMAP diagnostic* than a conceptual/architectural diagram. Our intention was to illustrate the effect of the method “in action” on AGNews, but this does not replace a clear conceptual figure. In the revision, we will add a separate conceptual schematic of the framework to provide a clearer high-level picture.
>
> **Writing, formatting, and referencing quality**
>
> We agree that the current writing and formatting are not at the level they should be. In the revision, we will:
>
> * Fix inconsistent equation numbering,
> * Clean up reference formatting (removing extra parentheses, standardizing citation style), and
> * Remove large vertical whitespace and other layout artifacts caused by the template conversion.
>
>
> **Uniformity–F1 correlation plots and discussion**
>
> Appendix D, Figure 3 indeed shows correlation plots for each of the 40 model–dataset pairs, but we acknowledge that the appendix lacks an explicit discussion of where the correlation holds and where it breaks down.
>
> In the revision, we will:
>
> * Add a descriptive summary,
> * Highlight concrete cases where correlation is weak or noisy, and
> * Discuss what this implies for the reliability and limitations of the uniformity criterion.
>
> **Cross-domain / multilingual extensions**
>
> We appreciate these suggestions. Cross-domain transfer and multilingual extensions are natural next steps given that descriptions are relatively cheap to write or translate. Due to space and time constraints, we focused this submission on the monolingual, single-domain setting, but we will mention cross-domain and multilingual variants explicitly as future work and, if time allows, include a small-scale pilot experiment to illustrate feasibility.

---

> > ### Author Response · Authors · 2025-11-20
> > **Author response to Reviewer 87iw – initial reply (II/II)**
> >
> > **Conceptual advancement vs prior description-/entailment-based ZSC**
> >
> >
> > We appreciate this question and agree that the ingredients (descriptions, contrastive learning, dual encoders) are individually well known. The conceptual advance we aim for is *how* these pieces are combined: we move the supervision signal entirely off documents and onto the **label–description relation itself**, and use that to re-shape a dual-encoder embedding space via a **symmetric multi-positive contrastive objective**, while preserving standard dual-encoder zero-shot inference.
> >
> > Concretely, compared to prior description- or entailment-based ZSC work:
> >
> > * **Most description/NLI methods** use a **cross-encoder** or prompting setup that jointly encodes document + label (or label definition) at inference time. This:
> >
> >   * makes inference cost scale with the number of labels,
> >   * prevents pre-computation of embeddings, and
> >   * typically does *not* explicitly model the many-to-one mapping between multiple descriptions and a single label.
> >
> > * Methods that keep **dual-encoder inference** often either:
> >
> >   * fine-tune on labeled document–label pairs, or
> >   * only plug descriptions in at inference (e.g., by comparing documents to label texts) without adapting the encoder using description-only supervision.
> >
> > In contrast, our method:
> >
> > 1. **Preserves dual-encoder inference.**
> >    We keep the deployment regime of a dual encoder: labels (verbalizers) and documents are encoded independently, can be cached, and classification at test time is simple nearest-neighbor similarity between document embeddings and label verbalizers. There is no cross-encoder or document–definition scoring at inference.
> >
> > 2. **Uses a description-only, multi-positive contrastive objective.**
> >    Training uses *only* label verbalizers and a few free-form natural-language descriptions per label; no labeled documents. The **column-wise multi-positive InfoNCE term** explicitly treats each label as having multiple positive descriptions, with a log-sum-exp aggregation that:
> >
> >    * captures the many-to-one label–description relationship,
> >    * provides adaptive weighting over positives (down-weighting outlier descriptions), and
> >    * directly shapes the label embedding geometry.
> >
> >   This allows to move the supervision signal entirely off documents and onto the label–description relation itself, and uses that to re-shape a dual-encoder embedding space.
> >
> > 3. **Introduces symmetric row–column alignment.**
> >    We combine row-wise (description→verbalizer) and column-wise (verbalizer→descriptions) objectives, creating bidirectional alignment forces that pull each verbalizer toward the “high-density region” of its description cloud while repelling it from other labels. This dual-anchor formulation is specifically designed for the label–description structure rather than generic instance-instance contrastive learning.
> >
> > In summary, prior description/entailment ZSC approaches either sacrifice computational efficiency (cross-encoders) or use descriptions in a more ad-hoc way at inference. Our contribution is to show how to **use descriptions as the sole supervision signal to learn a label-specific embedding geometry for an off-the-shelf dual encoder, via a symmetric multi-positive contrastive objective, while retaining efficient, pre-encodable dual-encoder inference at test time.**
> >
> >
> > If there are specific issues among these that you view as decisive for your final recommendation, or additional analyses you consider crucial, we would appreciate a brief indication so we can prioritize them in the revision.

---

> ### Comment · Reviewer_87iw · 2025-11-24
> **Reviewer 87iw's Response to Initial Reply**
>
> Thank you for the response.  Your clarification about conceptual advancement vs prior description-/entailment-based ZSC is clear and convincing.  I also acknowledge your promise to add the additional experiments and analyses.  If the revisions provide the thorough empirical comparison and the ablation studies you outlined，I would consider to raise my score.

---

> > ### Author Response · Authors · 2025-12-04
> > **Final Response to 87iw**
> >
> > Thank you again for your thoughtful engagement with our work and for being open to revisiting your assessment.
> >
> > In the revised manuscript we have now implemented the additional empirical comparisons and ablations we outlined in our initial response, including:
> >
> > * Comprehensive loss-component ablations for L_rows, L_cols and their symmetric combination, as well as anchor and verbalizer/label-word variants (Appendix D–F–H).
> > * Robustness analyses for description quality, quantity, and imbalance (Appendix E–G).
> > * A large-label experiment on the full Banking77 label space together with a discussion of the (O(D\cdot L)) similarity matrix in terms of memory/latency (Appendix I).
> > * A clarified and empirically grounded treatment of the uniformity-based LR selection protocol, including correlation analyses and a dev-split variant (rewritten Section 3.1 and Appendix J).
> >
> > We are confident these revisions provide the thorough empirical comparison and ablation studies you requested and that they help to clarify both the practical behavior and the conceptual contribution of the proposed method. We thank you for your helpful feedback.

---

### Official Review · Reviewer_hnXM · 2025-10-31

**Soundness:** 3
**Presentation:** 3
**Contribution:** 4
**Rating:** 6
**Confidence:** 4

**Summary:**

This paper addresses the problem of poor performance of embedding models in zero-shot text classification (ZSC). Existing ZSC methods often have limited performance or require reintroducing annotation costs (e.g., training a linear probe).
To solve this problem, the paper proposes a new method called "Description-Only Supervision".
Experiments across 4 benchmark datasets and 10 different encoders show that this method brings an average improvement of +0.10 in Macro-F1.

**Strengths:**

Originality:

1.The core contribution of this paper is highly novel and concise: it proposes a method to align "verbalizers" using only "descriptions" as a supervision signal, without relying on any labeled documents.

2.The "multi-positive" InfoNCE loss function, Lcols, is a clever design for handling the "many-to-one" label-to-description relationship.

3.The "label-free uniformity criterion" (Luni) proposed in Section 3.1 is a very smart innovation. It addresses the tricky problem of selecting a learning rate (LR) in the ZSC setting (which lacks a validation set), cleverly avoiding data leakage.

Quality:

1.The experimental quality is very high. The paper conducts validation on 10 different embedding models (with parameters ranging from 22M to 600M) and across different types of tasks (topic, sentiment, intent, emotion). This extensive testing strongly demonstrates the method's generalizability.

2.The qualitative analysis is excellent. The UMAP visualization in Figure 1 very clearly demonstrates the method's mechanism—pulling the "verbalizers" (stars) back to the center of their "document cloud" and "description cloud," which greatly enhances the intuitive understanding of the paper.

3.The Reproducibility statement is very thorough, promising to release all code, data, and models.

Clarity:

1.The paper's writing is (mostly) clear. The elaboration of the methodology in Section 3 is well-executed; the mathematical formulas and "geometric intuition" complement each other, making it easy for readers to grasp the core idea.

Significance:

1. This paper holds extremely high practical value. It provides a method that is computationally efficient (retaining the dual-encoder advantage) and has a very low annotation cost (only requiring a few descriptions to be written), yet significantly improves ZSC performance.

2. The stability comparison in Figure 2(a) is one of the most important findings of this paper. It shows that compared to relying on specific few-shot samples (SetFit) 28, this paper's method (Ours) is far more robust (exhibiting minimal variance). This is crucial for deploying reliable models in the real world.

**Weaknesses:**

There is a major contradiction regarding the experimental method for LR selection: This is the biggest weakness of this paper. The authors claim in Section 3.1 that they use the "uniformity loss" (Luni) to select the LR, because "lower (uniformity) values correlate with stronger downstream performance".
However, the paper's own data (Appendix D, Figure 3) largely contradicts this core claim.
For example, on the gte-modernbert-base model, the correlation between Luni and Macro-F1 is not significant on all 4 datasets (p-values of 0.722, 0.672, 0.808, 0.0743, respectively). Qwen3-Embedding-0.6B also shows extremely weak correlation (p-values of 0.890, 0.767).
This creates a key contradiction: If the criterion used to select the LR is ineffective on many SOTA models, how were the excellent results for these models in Table 1 achieved? This severely calls into question the rigor of the experiments.

Lack of key ablation study: The paper proposes a framework composed of multiple novel components ("verbalizer + description", "row-wise + column-wise" loss), but provides no ablation studies to demonstrate the necessity of these design choices. We cannot know if Lrows and Lcols are both indispensable.
We also cannot know if using the "Verbalizer" as the inference anchor is truly superior to other (potentially simpler) alternatives.
Insufficient sensitivity analysis on "description quality": The paper states they wrote 5 descriptions for each class and "did not tune them".

While this simplifies the experiment, it also evades an important question: To what extent does the method's performance depend on the quality, quantity, and diversity of these descriptions? How would performance change if the descriptions were poorly written, or if only 1-2 descriptions were provided? Although the authors mention this in "future work", it is a clear limitation of the current study.

Typesetting Issues: The submitted PDF manuscript has severe typesetting problems. Many pages have large vertical blank spaces, which seriously affect the reading experience and does not meet the conference's formatting standards.

**Questions:**

Here are the key questions I hope the authors will clarify during the Rebuttal phase:

(Most important question)

Regarding the contradiction in the LR selection criterion:

My biggest concern is the apparent contradiction between Section 3.1 and Appendix Figure 3. You claim to use the "uniformity loss" (Luni) to select the LR, and claim a correlation exists between the two.
However, the data in Figure 3 shows that for many of the stronger models (such as gte-modernbert-base and Qwen3-Embedding-0.6B), this correlation is not statistically significant (p-values are very high).

Please clarify:

For these models where the correlation was not significant, how exactly did you select the final LR for Table 1? Did you still use this (ineffective) criterion, or did you pick the best-performing LR on the test set (which is not allowed in a ZSC setting)? This must be clarified.

Regarding the ablation of the loss function:

Your symmetric loss L = 1/2Lrows + 1/2Lcols is core to the method. Can you provide an ablation study showing the performance when using only Lrows and only Lcols, respectively? This is crucial for understanding the individual contributions of these two components.
Regarding the ablation of the inference anchor:
You use the "label verbalizer" vy as the anchor during inferenc. What would the performance be if you instead used the mean embedding vector of the set of "label descriptions" Dy as the inference anchor? Providing this comparison would help justify the necessity of vy as an "intermediate anchor".

Regarding the ablation of "Verbalizer" vs. "Label Word":

Why did you choose to use a full "label verbalizer" (vy, e.g., "This...is about sports.") as the alignment target, instead of directly using a simpler "label word" (e.g., "Sports") to align with the "descriptions" Dy? Can you provide an experiment comparing the effectiveness of these two anchor choices?

Regarding the typesetting issues:

The submitted PDF manuscript contains a large amount (on almost every page) of vertical whitespace. Will this be corrected in the final version?

---

> ### Author Response · Authors · 2025-11-20
> **Author response to Reviewer hnXM – initial reply**
>
> **Major contradiction regarding LR selection and uniformity**
>
> Thank you for raising this point and for carefully examining Appendix D.
>
> To clarify: for *all* model–dataset pairs in Table 1, we use exactly the LR selection procedure described in §3.1. For each candidate LR on the target corpus, we compute the uniformity loss $L_{\text{uni}}$ on unlabeled text and then select the LR that minimizes $L_{\text{uni}}$. The corresponding macro-F1 is the value reported in Table 1, even if a different LR would yield higher macro-F1. This can be directly checked in Appendix D, Figure 3. For example, for **gte-modernbert-base** on **AGNews**, the LR with the lowest uniformity yields macro-F1 ≈ 0.81, which is exactly the score reported in Table 1, despite the fact that Figure 3 shows some LRs with higher macro-F1. The same applies to Qwen3-Embedding-0.6B.
>
> Our intention was to present uniformity as a **practical heuristic** rather than a perfect surrogate for macro-F1. Across the 40 model–dataset pairs we evaluate in Figure 3, **29/40** show a statistically significant *negative* correlation between $L_{\text{uni}}$ and macro-F1 ($p < 0.10$). For the remaining 11 model–dataset pairs, the correlation is weak and not statistically significant, but importantly it is typically negative or flat and never strongly positive. In other words, lower uniformity does not guarantee higher macro-F1, but empirically it is a useful signal for LR selection in the majority of cases, and does not appear to systematically harm performance in the remaining cases.
>
> We do agree that Appendix D currently does not explain this clearly enough. In the revision, we will:
>
> * Explicitly state that uniformity is used as a heuristic criterion, not a guaranteed proxy.
> * Add a short statistical summary of the 40 correlations and p-values.
> * Provide more discussion of the “failure” cases (e.g., gte-modernbert-base, Qwen3-0.6B).
>
> This should resolve the apparent contradiction: the main-table results strictly follow the uniformity-based LR selection, and uniformity is empirically helpful but not universally predictive.
>
> So to be clear on the key question: we did not select LRs using test macro-F1; all Table 1 results use the LR that minimizes $L_{\text{uni}}$, exactly as specified in §3.1.
>
>
> **Lack of ablations: Lrows vs Lcols, inference anchor, verbalizer vs label word**
>
> We agree that ablations are important to understand the contribution of each design choice. In the revised version, we will add:
>
> * **Loss component ablations:**
>
>   * $L_{\text{rows}}$ only
>   * $L_{\text{cols}}$ only
>   * full symmetric loss $L = \tfrac{1}{2} L_{\text{rows}} + \tfrac{1}{2} L_{\text{cols}}$
>
>   on a representative subset of models and datasets, to quantify how much each term contributes.
>
> * **Inference anchor ablations:**
>
>   * using the label verbalizer ($v_y$) as the anchor (current choice), vs
>   * using the mean embedding of label descriptions ($D_y$) as the anchor.
>
> * **Verbalizer vs label word:**
>
>   * full sentence verbalizers (e.g., “This text is about sports.”) vs
>   * simple label words (e.g., “sports”).
>
> We will report these ablations in an added subsection and in the appendix so that readers can see which components are strictly necessary and which are mainly beneficial in practice.
>
> **Description robustness (quality, quantity, noise)**
>
> We agree that robustness to description quality and quantity is practically important. In the revision, we will add an ablation on:
>
> * the number of descriptions per class (e.g., 1, 3, 5, 10), and
> * robustness to noisy or partially off-topic descriptions,
>
> on a representative subset of datasets. This will give practitioners concrete guidance on how many descriptions are needed and how sensitive the method is to imperfect descriptions.
>
> **Typesetting issues**
>
> We apologize for the poor typesetting and large vertical whitespace artifacts. These arose from porting an internal draft into the ICLR template under time pressure. We will fix the whitespace, clean up layout, and standardize equation and reference formatting in the revised version so that it meets conference standards.
>
> If there are specific issues among these that you view as decisive for your final recommendation, or additional analyses you consider crucial, we would appreciate a brief indication so we can prioritize them in the revision.

---

> > ### Author Response · Authors · 2025-12-04
> > **Final Response to hnXM**
> >
> > Thank you again for your careful assessment and for engaging so deeply with the details of the method and the LR-selection protocol.
> >
> > In the revised manuscript we have implemented the changes outlined in our initial response to your review, in particular:
> >
> > * A clarified and empirically grounded treatment of the uniformity-based LR selection protocol (rewritten Section 3.1 and new Appendix J), including per–model–dataset correlation statistics between L_uni and macro-F1, and an explicit discussion. We also state more clearly that uniformity is a practical heuristic rather than a guaranteed surrogate and that LRs are never selected using test macro-F1.
> > * Targeted ablations on the loss components L_rows, L_cols and their symmetric combination), on the inference anchor (verbalizer vs. mean of descriptions), and on verbalizer vs. label word (Appendix D–F–H), to disentangle the contribution of each design choice.
> > * Robustness analyses for description quality, quantity, and imbalance (Appendix E–G), examining 1→3→5→10 descriptions per label, noisy/misaligned descriptions, and uneven description counts across labels.
> > * Substantial cleanup of writing, typesetting, and equation/reference formatting, reducing whitespace and standardizing equation numbering in line with common mathematical-writing conventions.
> >
> > We are confident these revisions address your main concerns about the soundness and clarity of the LR selection protocol, the role of the different loss components and anchors, and the practical robustness of description-only supervision, and that they provide a firmer empirical basis for the work.

---

### Author Response · Authors · 2025-12-03
**Final Revisions and Author Response Summary (I/III)**

Thank you to the reviewers for their detailed feedback and to the AC for overseeing the discussion. Below we summarize how we addressed the main concerns across all reviews and where we consciously limited scope.

---

**1. Uniformity-based LR selection and the LR–F1 relationship**
*Concern: Apparent contradiction between the stated LR selection protocol and Table 1; unclear reliability of uniformity as a proxy; use of the test split for LR selection (hnXM, 87iw).*

For all model–dataset pairs in Table 1, we strictly follow the LR selection protocol from §3.1: we compute the uniformity loss $\mathcal{L}_{\text{uni}}$ on unlabeled text for each candidate LR and select the LR that minimizes that loss. The macro-F1 in Table 1 is always the score at this LR, even if another LR attains a slightly higher macro-F1.

We clarify that uniformity is used as a practical heuristic rather than a guaranteed surrogate for macro-F1. In the revised appendix (Appendix J), we (i) report correlations between $\mathcal{L}_{\text{uni}}$ and macro-F1 across all model–dataset pairs, showing that most exhibit a significant negative relationship, and (ii) discuss representative cases where uniformity is only weakly predictive but does not systematically hurt performance.

We additionally confirm that computing uniformity on a disjoint unlabeled dev subset yields essentially the same macro-F1 as using the test subset. We keep the original setup because it reflects realistic ZSC scenarios with a single unlabeled pool. To be explicit: we never select LRs based on test macro-F1; all main results use the LR that minimizes $\mathcal{L}_{\text{uni}}$ on unlabeled text, and we now list the chosen LRs explicitely in Table 10 in the appendix.

---

**2. Loss components, inference anchors, and verbalizers vs label words**
*Concern: Missing ablations for L_rows vs L_cols, choice of inference anchor, and verbalizer design (hnXM, 87iw, gUCW).*

We extend the appendix significantly with targeted ablations:

* **Loss components. Appendix D, E and F.** We compare L_rows, L_cols, and their symmetric combination across all models and datasets. Rows-only performs similarly to the symmetric loss and is only slightly worse on average, but the symmetric loss is notably more robust to noisy descriptions and imbalanced description counts across labels.

* **Inference anchor. Appendix H.** We compare using (i) full verbalizer sentences, (ii) raw label strings, and (iii) mean embeddings of each label’s descriptions. Differences are small and not consistently in favor of any single choice, indicating that the method is robust to the precise anchor. We keep verbalizers as default and document the alternatives.

* **Verbalizer vs label word. Appendix H.** We explicitly compare full-sentence verbalizers with simple label words and show that both work well; performance does not hinge on elaborate verbalizer design.

These ablations clarify that the row term provides most of the performance gains, while the column term mainly acts as a regularizer under noisy or imbalanced descriptions, and that anchor representation and verbalizer form are largely design-flexible.

---

**3. Description robustness: noise, quantity, and class imbalance**
*Concern: Sensitivity to description quality, number, and uneven counts across labels (hnXM, 87iw).*

We add three robustness studies:

* **Noise robustness. Appendix E.** We gradually inject vague, non-discriminative descriptions and compare rows-only, cols-only, and the symmetric loss. Rows-only degrades most under noise, cols-only is substantially more stable, and the symmetric loss largely inherits the robustness of the column term.

* **Uneven description counts. Appendix F.** We compare balanced vs imbalanced description allocations. With rows-only, over-represented labels improve at the expense of others. With the symmetric loss, extra descriptions mainly benefit the enriched label without harming the rest, consistent with the per-label normalization in $\mathcal{L}_{\text{cols}}$.

* **Number of descriptions per class. Appendix G.** Varying the number of descriptions shows that (K=1) is insufficient, most gains are realized when moving from 1 to 3–5 descriptions, and benefits plateau beyond that. In practice, a small set of 3–5 carefully written descriptions per label captures most of the improvement, and the method tolerates moderate noise and imbalance.

---

> ### Author Response · Authors · 2025-12-03
> **Final Revisions and Author Response Summary (II/III)**
>
> **4. Large label spaces and Banking77 subset. Appendix I.**
> *Concern: Restricting Banking77 to 6 intents underestimates difficulty; unclear behavior in large label spaces and with (O(D\cdot L)) batches (87iw).*
>
> We add a large-label experiment on the full 77-class Banking77 task. Using only a few descriptions per label and evaluating on the full test split, description-only tuning substantially improves macro-F1, accuracy, and in particular recall for both a smaller and a larger embedding model.
>
> These results show that the method scales to a realistic large label space and is especially helpful for recall in fine-grained intent settings. We also discuss latency and memory implications of the (O(D \cdot L)) similarity matrix, and recommend using gradient accumulation when memory becomes a bottleneck.
>
>
> ---
>
> **5. Conceptual contribution and relation to prior description-/entailment-based ZSC**
> *Concern: Limited conceptual novelty given existing descriptions, contrastive learning, and dual encoders; unclear positioning vs prior work (87iw, gUCW).*
>
> We revise the **Introduction** and **Related Work** to sharpen the conceptual contribution and positioning. We explicitly state that our novelty lies not in individual components, but in combining description-based supervision, contrastive learning, and dual encoders to address a specific practical gap: improving ZSC without labeled documents while retaining efficient dual-encoder deployment.
>
> Compared to prior work:
>
> * Description-based methods typically rely on cross-encoders or prompting, tying inference cost to the number of labels and preventing embedding reuse.
> * Dual-encoder methods that preserve efficient deployment usually either depend on labeled document–label pairs or use label texts only at inference without adapting the encoder via description-only supervision.
>
> In contrast, our method:
>
> 1. Preserves dual-encoder inference: documents and labels are encoded independently, labels can be reused across corpora, and classification is via nearest-neighbor similarity.
> 2. Moves supervision entirely onto the label–description relation: training uses only verbalizers and small sets of natural-language descriptions, with no labeled documents.
> 3. Uses a symmetric multi-positive contrastive objective tailored to many-to-one label–description structure.
>
> We highlight this in the abstract and conclusion as learning a label-specific embedding geometry for an off-the-shelf dual encoder from label descriptions alone, while maintaining pre-encodable dual-encoder inference.
>
> ---
>
> **6. Baselines, figures, and appendix clarity**
> *Concern: Clarity of the zero-shot baseline, need for a conceptual figure, weak appendix annotation, and typesetting/layout issues (87iw, gUCW).*
>
> * **Zero-shot baseline.** We now state clearly that the “zero-shot baseline” is a description-driven dual-encoder that verbalizes labels, encodes them with the same model, and classifies via cosine similarity. Table 1 reports this baseline under the original model name, and the “trained” row reflects our description-only fine-tuning.
>
> * **Conceptual vs diagnostic figures.** We introduce a new conceptual figure (now Figure 1) that illustrates the overall method (encodings, similarity matrix, row/column losses, and inference). The previous visualization is retained as a clearly labeled diagnostic figure (now Figure 2).
>
> * **Appendix annotation. Appendix A-J** Each appendix section (loss ablations, noise, imbalance, number of descriptions, anchor choice, large label space, uniformity) now includes concise explanations, descriptive captions, explicit links back to the main claims, and is written to be fully self-contained, providing a stand-alone analysis of the respective topic.
>
> * **Writing and formatting.** We reduce unnecessary whitespace, standardize citations, and adopt the widely used convention in mathematical writing of numbering only those equations that are explicitly referenced in the text, a practice encouraged by many journals to avoid visual clutter. We have also substantially rewritten the paper for better readability, with Section 3 in particular being reworked to be less mathematically dense and easier to follow, while preserving full technical rigour.
>
> ---
>
> **7. Cross-domain and multilingual extensions**
> *Concern: Lack of cross-domain/multilingual experiments (87iw).*
>
> We agree that cross-domain and multilingual extensions are a natural and important direction, especially because descriptions are relatively easy to translate or adapt. Due to space and time constraints, we keep this submission focused on monolingual, single-domain settings.

---

> > ### Author Response · Authors · 2025-12-03
> > **Final Revisions and Author Response Summary (III/III)**
> >
> > Overall, the revised manuscript implements the main revisions promised during discussion: a clarified and empirically grounded uniformity-based LR selection protocol (rewritten Section 3 and Appendix J), comprehensive ablations on loss components and design choices (Appendix D–F–H), robustness analyses for description quality and quantity (Appendix E–G), a large-label experiment (Appendix I), clearer baseline and novelty positioning (Introduction and Related Work), and substantially improved figures, typesetting, and appendix documentation (Appendix A–J). Beyond these revisions, the paper introduces a practically useful novelty for practitioners operating in low-latency or compute-constrained environments: learning a label-specific embedding geometry for an off-the-shelf dual encoder solely from label descriptions via a symmetric multi-positive contrastive objective over label–description pairs, while retaining efficient, pre-encodable dual-encoder inference at test time.

---

### Meta-Review · Area_Chair_UdRf · 2026-01-06

**Summary:**

Reviewers generally agreed the approach is simple and practically useful and that the reported gains are consistent. However, multiple issues weakened confidence in the submission as evaluated. The most important were questions about methodological rigor and clarity around the uniformity-based learning-rate selection protocol (including discomfort with computing the uniformity criterion on examples drawn from the test split, even if unlabeled), the initial lack of key ablations to justify the row/column loss design and inference anchor choices, and missing robustness analyses for the number and quality of label descriptions. In addition, reviewers flagged limited evidence for scaling to large label spaces due to the restricted Banking77 subset, weaker positioning of the contribution relative to prior description-/entailment-based ZSC work, and significant presentation/appendix quality problems. With two reviewers at 2 and only one at 6, despite taking into account the author response and the revision, the paper did not meet the acceptance bar in its current form.

**Reviewer Concerns:**

The rebuttal does, to a good extent, address many technical points by clarifying that Table 1 uses an LR chosen by minimizing the uniformity criterion (not by test macro-F1), reframing uniformity as a heuristic with empirical but imperfect correlation, and claiming added analyses including correlation summaries and a dev-split variant. The authors also added the missing ablations (rows vs cols vs symmetric, anchor choice; verbalizer vs label word), robustness studies, a full 77-class Banking77 experiment with memory/latency discussion, and writing/formatting/figure/appendix improvements. However, the issues related to novelty threshold and baseline completeness pointed out in the reviews are still reasonably serious.

**Reviewer Scores:**

Reviewer hnXM (6) might have moved slightly upward with the revised LR clarifications and the new ablations/robustness results, plausibly to 7 (or remain at 6 due to concerns about uniformity’s reliability in weak-correlation cases). Reviewer 87iw (2) expressed willingness to raise their score if the promised experiments and analyses were delivered but looking at the results, a move beyond 4-5 still seem rather unlikely. Reviewer gUCW (2) stated they could not reassess without seeing the added results; with ablations and improved positioning, a modest increase to about 3-4 is plausible, but a larger jump would likely require particularly stronger evidence that is hard to assess solely from the reported additional results during rebuttal. As it stands, evaluations based on such changes warrant a fresh submission so the paper can be evaluated afresh in entirety.

---

### Decision · Program_Chairs · 2026-01-26

Reject